# Sequentially Auditing Differential Privacy

**Tomás González**
Carnegie Mellon University
`tcgonzal@andrew.cmu.edu`

**Mateo Dulce Rubio**
New York Universiy
`mateo.d@nyu.edu`

**Aaditya Ramdas**
Carnegie Mellon University
`aramdas@cs.cmu.edu`

**Mónica Ribero**
Google Research
`mribero@google.com`

## Abstract

We propose a practical sequential test for auditing differential privacy guarantees of black-box mechanisms. The test processes streams of mechanisms' outputs providing anytime-valid inference while controlling Type I error, overcoming the fixed sample size limitation of previous batch auditing methods. Experiments show this test detects violations with sample sizes that are orders of magnitude smaller than existing methods, reducing this number from 50K to a few hundred examples, across diverse realistic mechanisms. Notably, it identifies DP-SGD privacy violations in *under* one training run, unlike prior methods needing full model training.

## 1 Introduction

Auditing privacy guarantees of algorithms that process sensitive data, prevalent in domains like finance, healthcare, and users' behavior on the web, is crucial for building and deploying reliable and trustworthy software. While rigorous privacy protections like Differential Privacy (DP) [15] have been adopted by industry [3, 20, 41] and government agencies [48], the promised theoretical guarantees rely on both correct algorithmic design (e.g., avoiding flaws or errors in mathematical proofs, like those demonstrated with the Sparse Vector Technique [30]) and correct implementation, where subtle bugs can compromise privacy (e.g., missing constants, incorrect sampling, or fixed seeds). Furthermore, theoretical privacy parameters $\varepsilon$ and $\delta$ often represent worst-case upper bounds, motivating the need for empirical privacy auditing to assess practical privacy leakage and gain intuition about algorithms' vulnerabilities [33].

Auditing privacy can be framed as a statistical hypothesis test: distinguishing the null hypothesis $H_0$: "the algorithm satisfies a claimed privacy guarantee", from the alternative $H_1$: "the algorithm violates the claimed privacy guarantee". Current approaches to this testing problem generally fall into two categories, each with significant drawbacks. First, parametric tests can achieve high statistical power but rely on strong, often unverifiable, assumptions about the specific workings and output distribution of the mechanism or the specific flaw [33, 16]. Second, black-box tests, where the auditor observes only the mechanism's outputs without knowledge of its internal functioning, make fewer assumptions but requires an unknown and typically large number of samples to draw statistically meaningful conclusions [5, 26]. This makes them impractical for auditing complex, resource-intensive algorithms such as differentially private stochastic gradient descent (DP-SGD), where obtaining even one sample often requires significant computation (e.g., backpropagation through millions of model parameters).

In this paper, we propose a new framework for approximate DP auditing that leverages recent advances in sequential hypothesis testing to overcome these limitations. Our approach uses e-values

[39], non-negative random variables that (when multiplied) *sequentially* accumulate evidence against the null hypothesis. Under the null $H_0$, the expectation of an e-value is bounded by 1; under the alternative $H_1$, well-designed (products of) e-values can grow exponentially fast, allowing for early stopping and efficient detection of privacy violations. This sequential methodology allows testing to proceed adaptively: samples are collected and evaluated iteratively, and the test stops as soon as a significance level $\alpha$ is reached. Notably, sequential tests automatically adapt to the unknown sample complexity of the problem, eliminating the need to fix the test's sample size a priori and avoiding unnecessary computation.

Our specific test statistics are built upon the Maximum Mean Discrepancy (MMD) [19], a powerful kernel-based metric for comparing probability distributions. MMD is particularly well-suited for the two-sample tests emerging in privacy auditing, as bounds on MMD can be easily translated to approximate DP parameters [26]. Further, MMD is highly flexible. It can incorporate prior knowledge in white-box settings—with access to intermediate gradients or the noise distribution family—or, as our experiments show, operate effectively in a black-box manner.

The main contributions of this work are summarized as follows:

- We introduce a new general MMD-based one-sided sequential testing framework. As a key application, we instantiate this framework for auditing approximate DP mechanisms (see Section 3). Our method enables anytime valid inference while controlling Type I error, eliminating the need for pre-defined test sample sizes, and ensuring a bounded expected stopping time under the alternative. We establish these theoretical guarantees in Theorems 3.2 and 3.3.

- We present a new connection between MMD and Hockey-Stick divergence in Theorem 3.1 that allows for the translation of MMD-based hypothesis tests to approximate DP auditing tests. The connection presented is tighter than previous bounds, increasing test power. Our experiments show that previous tests requiring hundreds of thousands of samples while still failing to reject, now reject with under a thousand samples.

- We validate our methods on common DP mechanisms with Gaussian and Laplace noise. We further demonstrate efficacy on auditing benchmark algorithms [26, 5] and provide results for the challenging case of DP-SGD [1, 46], showcasing the practical benefits of early failure detection enabled by our sequential approach.

**Related work.**     At the core of approximate DP auditing is the estimation of the effective privacy loss from samples of the mechanism. There is substantial prior work on this, primarily situated within the batch setting, where a fixed number of samples are drawn [18, 12, 14, 11, 5, 35, 28, 29]. Also in the batch setting, a significant body of work specializes on auditing machine learning models, particularly those trained with DP-SGD. This includes membership inference attacks (MIA) [24, 37, 7, 23, 33] and data reconstruction attacks [21, 4, 31].

Our approach relies on sequential testing with kernel methods, specifically the Maximum Mean Discrepancy (MMD) [19], to construct test statistics. Prior work on privacy auditing has also used kernel methods, such as estimating regularized kernel Rényi divergence [13], but often requires strong assumptions (e.g., knowledge of covariance matrices) impractical in black-box settings or for mechanisms beyond Gaussian or Laplace.

To the best of our knowledge, applying modern sequential hypothesis testing techniques, particularly those based on e-values or test supermartingales [43, 38], to the general problem of black-box DP auditing is novel. However, sequential testing by betting ideas have been successfully applied for auditing in other settings like elections [52], finance [45], fairness [8], and language models [40]. The most closely related work is [40] as they also propose a one-sided test, although for detecting distribution shifts in language models. While their setting is similar, their testing procedure differs from ours in key ways, allowing us to establish results they do not—such as bounds on the expected number of samples until rejection under the alternative. We elaborate on these differences in Section 3.2.

In Appendix H we discuss in detail more references and how they compare to our work.

## 2 Preliminaries

**Notation.** Throughout the paper we let $\mathcal{D} \subseteq \cup_{d \in \mathbb{N}} \mathbb{R}^{p \times d}$ be a set of datasets; datasets $D \in \mathcal{D}$ have a finite but arbitrary number of $p$-dimensional records. We denote by $\mathcal{A} : \mathcal{D} \mapsto \mathcal{X}$ randomized mechanisms that map datasets to a range $\mathcal{X}$. We say that $S, S' \in \mathcal{D}$ are neighboring datasets (denoted by $S \sim S'$) if they differ by at most one data point. While our framework is agnostic to the definition of "neighboring", for simplicity (and in our experiments) we assume $S \sim S'$ in the add/remove framework, where $S'$ can be obtained by adding or removing exactly one record from $S$. Given a reproducing kernel Hilbert space (RKHS) $\mathcal{H}$, let $0_{\mathcal{H}} \in \mathcal{H}$ be the constant zero function.

**Definition 2.1.** *The Hockey-Stick divergence between $P$ and $Q$ (of order $e^{\varepsilon}$) is the $f$-divergence with $f(x) = \max\{0, x - e^{\varepsilon}\}$. Specifically,*

$$D_{e^{\varepsilon}}(P \| Q) := \mathbb{E}_Q \left[ f \left( \frac{dP}{dQ} \right) \right]. \tag{1}$$

**Definition 2.2** (**Approximate Differential Privacy** [15]). *A randomized algorithm $\mathcal{A} : \mathcal{D} \mapsto \mathcal{X}$ is $(\varepsilon, \delta)$-differentially private if for any pair of neighboring datasets $S$ and $S'$ and any event $\mathcal{E} \subseteq \mathcal{X}$, $\mathbb{P}[\mathcal{A}(S) \in \mathcal{E}] \leq e^{\varepsilon} \mathbb{P}[\mathcal{A}(S') \in \mathcal{E}] + \delta$. Equivalently, $D_{e^{\varepsilon}}(\mathcal{A}(S) \,\|\, \mathcal{A}(S')) \leq \delta$.*

The sequential tests we propose require estimating the *witness function* for a given divergence of interest. Intuitively, this witness function can be thought as the function that highlights the maximum difference between two distributions as measured by the underlying divergence. The definition of approximate DP uses the Hockey-Stick divergence, whose witness function is not very easy to work with as it can be highly non-smooth. Instead, we will use the MMD as a notion of the distance between the distributions $\mathcal{A}(S)$ and $\mathcal{A}(S')$, and leverage the connection of MMD to approximate DP introduced in Theorem 3.1.

**Definition 2.3.** *Let $\mathcal{H}$ be a reproducing kernel Hilbert space with domain $\mathcal{X}$ and kernel $K(\cdot, \cdot)$ such that $K(x, x) \leq 1$ for all $x \in \mathcal{X}$. Given two distributions $\mathbb{P}, \mathbb{Q}$ supported on $\mathcal{X}$, define*

$$\mathrm{MMD}(\mathbb{P}, \mathbb{Q}) := \sup_{\|f\|_{\mathcal{H}} \leq 1} \mathbb{E}_{X, Y \sim (\mathbb{P}, \mathbb{Q})}[f(X) - f(Y)].$$

*The solution $f^*$ achieving the supremum is called the witness function.*

Below, we introduce a definition and a theorem from the literature of supermartingales which are key to the design and analysis of our sequential tests.

**Definition 2.4** (Nonnegative supermartingale). *An integrable stochastic process $\{\mathcal{K}_t\}_{t \geq 0}$ is a nonnegative supermartingale if $\mathcal{K}_t \geq 0$ and $\mathbb{E}[\mathcal{K}_t \mid \mathcal{K}_1, ..., \mathcal{K}_{t-1}] \leq \mathcal{K}_{t-1}$, where the inequalities are meant in an almost sure sense.*

**Theorem 2.1** (Ville's inequality [49]). *For any nonnegative supermartingale $\{\mathcal{K}_t\}_{t \geq 0}$ and $\alpha > 0$, $\mathbb{P}[\exists t \geq 0 : \mathcal{K}_t \geq 1/\alpha] \leq \alpha \mathbb{E}[\mathcal{K}_0]$.*

## 3 Sequential Auditing

This section formalizes the proposed auditing by betting framework. We start by formally introducing the problem statement and continue by introducing an MMD-based test statistic. Subsequently, we develop Algorithm 1, a sequential test for privacy auditing. We then provide formal guarantees on controlling Type I error uniformly over the null, and demonstrating exponential growth under the alternative, which implies a finite stopping time when detecting faulty algorithms. We conclude this section with a comparative discussion of two subroutines within the main sequential algorithm.

Auditing DP can be seen as a two-step procedure: (1) identifying worst-case neighboring datasets $S \sim S'$ that maximize the discrepancy between the distributions $\mathcal{A}(S)$ and $\mathcal{A}(S')$, and (2) testing whether the privacy guarantee holds for $S \sim S'$ while controlling Type I error at level $\alpha$. In this work we focus on the testing problem; that is, throughout the remainder of this work we assume the pair $S \sim S'$ is fixed. While we do not require access to the worst-case pair $S \sim S'$, the statistical power of our test increases with larger $\mathrm{MMD}(\mathcal{A}(S), \mathcal{A}(S'))$ (see Theorem 3.3).

### 3.1 Problem statement

**Definition 3.1.** *(Hockey-Stick approximate DP test) We consider an auditor that employs a binary hypothesis test $\phi$ designed to evaluate whether an algorithm $\mathcal{A}$ satisfies the $(\varepsilon, \delta)$-approximate DP guarantee on neighboring datasets $S \sim S'$. The test has access to streams of i.i.d. observations $\mathbf{X} = (X_1, X_2, ...) \overset{iid}{\sim} \mathcal{A}(S), \mathbf{Y} = (Y_1, Y_2, ...) \overset{iid}{\sim} \mathcal{A}(S')$ from $\mathcal{A}$ evaluated on $S$ and $S'$, respectively. The goal of the test is to distinguish between the following two hypotheses:*

$$H_0: \quad D_{e^\varepsilon}(\mathcal{A}(S) \,||\, \mathcal{A}(S')) \leq \delta, \qquad H_1: \quad D_{e^\varepsilon}(\mathcal{A}(S) \,||\, \mathcal{A}(S')) > \delta.$$

*The auditor rejects $H_0$ when $\phi(\mathbf{X}, \mathbf{Y}) = 1$ and fails to reject it otherwise.*

The Hockey-Stick divergence relies on the likelihood ratio of the potentially unknown distributions $\mathcal{A}(S)$ and $\mathcal{A}(S')$; consequently, it can be very challenging in practice to develop high-power tests with a reasonable number of samples [26] for the test described in Definition 3.1. Theorem 3.1 below introduces a tighter connection between the Hockey-Stick divergence and the MMD compared to previous bounds [26]. This tighter connection enables the development of an MMD-based hypothesis test for approximate differential privacy. The improved bound inherently increases the power of kernel-MMD-based Hockey-Stick tests (see Appendix A).

**Theorem 3.1.** *(Improvement of [26] Theorem 4.13) Let $\mathcal{H}$ be a reproducing kernel Hilbert space with domain $\mathcal{X}$ and kernel $K(\cdot, \cdot)$ such that $0 \leq K(x, y) \leq 1$ for all $x, y \in \mathcal{X}$. If mechanism $\mathcal{A}$ is $(\varepsilon, \delta)$-DP, then for any $S \sim S'$,*

$$\mathrm{MMD}(\mathcal{A}(S), \mathcal{A}(S')) \leq \sqrt{2}\left(1 - \frac{2(1-\delta)}{1 + e^\varepsilon}\right). \tag{2}$$

Theorem 3.1 offers a strict improvement over [26, Theorem 4.13]. Further, the new bound has the nice property of not becoming vacuous as $\varepsilon$ increases. In fact, it approaches $\sqrt{2}$ as $\varepsilon \to \infty$, while the previous bound grows to infinity with $e^\varepsilon$.

Theorem 3.1 implies that if $\mathrm{MMD}(\mathcal{A}(S), \mathcal{A}(S')) > \sqrt{2}\left(1 - \frac{2(1-\delta)}{1+e^\varepsilon}\right)$, then $\mathcal{A}$ cannot be private. In light of this, we will focus on the following test, which is based on the MMD instead of the Hockey-Stick divergence.

**Definition 3.2.** *(MMD approximate DP test) Under the same conditions as Definition 3.1, let $\tau(\varepsilon, \delta) := \sqrt{2}\left(1 - \frac{2(1-\delta)}{1+e^\varepsilon}\right)$. We define the MMD-approximate DP test as the task of distinguishing between the following two hypotheses:*

$$H_0: \quad \mathrm{MMD}(\mathcal{A}(S) \,||\, \mathcal{A}(S')) \leq \tau(\varepsilon, \delta), \qquad H_1: \quad \mathrm{MMD}(\mathcal{A}(S) \,||\, \mathcal{A}(S')) > \tau(\varepsilon, \delta).$$

Previous work on DP auditing often focuses on the fixed batch sample size case that proceeds as follows: Take $n$ samples from $\mathcal{A}$ on each dataset to obtain samples $X_1, .., X_n \overset{iid}{\sim} \mathcal{A}(S)$ and $Y_1, ..., Y_n \overset{iid}{\sim} \mathcal{A}(S')$. Then use these samples to construct an $(1-\alpha)$-confidence interval for $D_{e^\varepsilon}(\mathcal{A}(S), \mathcal{A}(S'))$. If the interval doesn't intersect $[0, \delta]$, then we conclude with high probability that $\mathcal{A}$ is not $(\varepsilon, \delta)$-DP. This approach can be problematic: running $\mathcal{A}$ can be computationally expensive in practice, so choosing an $n$ that is too large can be infeasible or simply lead to wasted computational resources. Conversely, choosing an $n$ that is too small might result in an inconclusive test, and samples often cannot be reused without decrease in statistical power.

We begin by presenting an abstract, potentially impractical test to build intuition, and then introduce the proposed practical sequential auditing framework in Algorithm 1.

### 3.2 An abstract template for $(\varepsilon, \delta)$-DP sequential testing

**Theorem 3.2.** *Let $f^*$ be the witness function for $\mathrm{MMD}(\mathcal{A}(S) \,||\, \mathcal{A}(S'))$ (as defined in Definition 2.3). Given samples $\{X_t, Y_t\}_{t \geq 1} \overset{iid}{\sim} \mathcal{A}(S) \times \mathcal{A}(S')$ and a fixed hyperparameter $\lambda > 0$, define the stochastic process $\{\mathcal{K}_t^*(\lambda)\}_{t \geq 0}$ as follows:*

$$\mathcal{K}_0^*(\lambda) = 1, \quad \mathcal{K}_t^*(\lambda) = \mathcal{K}_{t-1}^*(\lambda) \times (1 + \lambda[f^*(X_t) - f^*(Y_t) - \tau(\varepsilon, \delta)]), \tag{3}$$

*where $\tau(\varepsilon, \delta)$, defined in Definition 3.2, represents the expected upper bound on MMD under the null hypothesis. Then it holds that:*

1. *Under the null hypothesis ($H_0 : \mathrm{MMD}(\mathcal{A}(S), \mathcal{A}(S')) \leq \tau(\varepsilon, \delta)$), for any $\lambda \in \left[0, \frac{1}{2+\tau(\varepsilon,\delta)}\right]$ and any $\alpha \in (0, 1)$, we have $\mathbb{P}[\sup_{t \geq 1} \mathcal{K}_t^*(\lambda) \geq 1/\alpha] \leq \alpha$.*

2. *Under the alternative ($H_1 : \mathrm{MMD}(\mathcal{A}(S), \mathcal{A}(S')) > \tau(\varepsilon, \delta)$), there exists a $\lambda^* \in \left[0, \frac{1}{8+4\tau(\varepsilon,\delta)}\right]$ such that $\lim_{t \to \infty} \frac{\log \mathcal{K}_t^*(\lambda^*)}{t} = \Omega(\Delta^2)$ almost surely, where $\Delta = \mathrm{MMD}(\mathcal{A}(S), \mathcal{A}(S')) - \tau(\varepsilon, \delta)$.*

Theorem 3.2 suggests an algorithm that sequentially accumulates evidence to reject hypothesis $H_0$ against $H_1$. Unfortunately, this requires to know $\lambda^*$ and the witness function $f^*$. In the following subsection, we show that $\lambda^*$ and $f^*$ can be learned: $\lambda^*$ using Online Newton Step (ONS), and $f^*$ from samples using Online Gradient Ascent (OGA); see Appendix E for more details on these algorithms.

Our approach builds upon the work in [44], and its application to our one-sided privacy auditing test requires substantial and non-trivial modifications. Specifically, the test proposed in [44] inherently addresses a two-sided test with hypothesis $H_0 : \mathrm{MMD}(\mathcal{A}(S), \mathcal{A}(S')) = 0$ versus $H_1 : \mathrm{MMD}(\mathcal{A}(S), \mathcal{A}(S')) > 0$. Since the MMD is non-negative by definition, this effectively corresponds to testing $H_0 : \mathrm{MMD}(\mathcal{A}(S), \mathcal{A}(S')) = 0$ against the two-sided alternative $H_1 : \mathrm{MMD}(\mathcal{A}(S), \mathcal{A}(S')) \neq 0$. In contrast, the privacy auditing setting needs a one-sided test against a specific, non-zero threshold. A natural extension would lead to testing $H_0 : \mathrm{MMD}(\mathcal{A}(S), \mathcal{A}(S')) = \tau(\varepsilon, \delta)$ versus $H_1 : \mathrm{MMD}(\mathcal{A}(S), \mathcal{A}(S')) \neq \tau(\varepsilon, \delta)$, which is different from the one-sided test with alternative $H_1 : \mathrm{MMD}(\mathcal{A}(S), \mathcal{A}(S')) > \tau(\varepsilon, \delta)$. This distinction is critical because, unlike the zero-threshold null hypothesis, values of MMD below $\tau(\varepsilon, \delta)$ are possible and relevant to the null hypothesis. Consequently, a simple extension of the two-sided framework would result in an inappropriate two-sided alternative $H_1 : \mathrm{MMD}(\mathcal{A}(S), \mathcal{A}(S')) \neq \tau(\varepsilon, \delta)$.

Designing this one-sided test introduces significant analytical challenges. Concretely, we have to constrain $\lambda$ to be nonnegative to preserve the supermartingale property of the stochastic process $\mathcal{K}_t^*$. This restriction reduces the space of admissible $\lambda$ values and changes the analysis needed to establish the existence of an optimal $\lambda^*$ that ensures that the process grows under the alternative hypothesis (part 2 of Theorem 3.2). We overcome this by proving the existence of such a $\lambda^*$ within the restricted domain, which required non-trivial adaptations and subtle but key modifications of the original results presented in [44]. This structural modification is essential to ensure the validity and power of the one-sided sequential test. Finally, while the framework in [44] is designed for a broader class of testing problems, our focused analysis of this specific two-sample MMD test allows for the derivation of clearer and more interpretable bounds. By carefully working out the specific constants for our problem, we avoid reliance on the more abstract machinery in their general framework.

We are not the first to draw inspiration from [44] to design a one-sided test. Recently, [40] proposed a one-sided test in the context of detecting behavioral shifts in language models for the hypotheses $H_0 : D(\mathbb{P}, \mathbb{Q}) \leq \tau$ versus $H_1 : D(\mathbb{P}, \mathbb{Q}) > \tau$ for some $\tau > 0$, where the distance function is defined as $D(\mathbb{P}, \mathbb{Q}) = \sup_{f \in \mathcal{F}} |\mathbb{E}_{X \sim \mathbb{P}, Y \sim \mathbb{Q}}[f(X) - f(Y)]|$ and $\mathcal{F}$ is a class of neural networks with scalar outputs in $[-1/2, 1/2]$. Their method also constructs a stochastic process that is a supermartingale under $H_0$ and can grow under $H_1$ under several assumption on the neural network class, but it differs from ours in two key respects. First, they consider a single stochastic process $\tilde{\mathcal{K}}_t$, while we define a family $\mathcal{K}_t^*(\lambda)$ parameterized by $\lambda$ (see eq. (3)). This parameterization enables the practical test we introduce in Section 3.3 to adaptively tune $\lambda$ and approximate the optimal growth rate. Second, even when $\lambda$ is fixed, our process exhibits faster expected growth under $H_1$. Specifically, if we were to adopt their approach for our one-sided test, we would obtain:[1]

$$\tilde{\mathcal{K}}_t = \prod_{i=1}^{t} \left( \frac{1 + \tau/2}{e^{\tau/2}} + \frac{1}{2} \cdot \frac{f^*(X_i) - f^*(Y_i) - \tau}{e^{\tau/2}} \right),$$

where we denote $\tau(\varepsilon, \delta)$ as $\tau$ for brevity. Under $H_1$, the expected value of this process satisfies:

$$\mathbb{E}_{H_1}[\tilde{\mathcal{K}}_t] = \left( \frac{1 + \tau/2}{e^{\tau/2}} + \frac{1}{2} \cdot \frac{\mathrm{MMD} - \tau}{e^{\tau/2}} \right)^t < \left( 1 + \frac{\mathrm{MMD} - \tau}{2 + \tau} \right)^t = \mathbb{E}_{H_1}\left[ \mathcal{K}_t^*\left( \frac{1}{2+\tau} \right) \right],$$

---

[1]Their process does not include the factors of $1/2$ from our expression; we include them to standardize both settings, as we work with function classes outputting in $[-1, 1]$ whereas theirs output in $[-1/2, 1/2]$.

where the inequality follows from $1 + \tau/2 < e^{\tau/2}$ for $\tau > 0$.

Finally, unlike our work, [40] does not analyze the expected growth of $\tilde{\mathcal{K}}_t$ under $H_1$, nor do they provide guarantees on the expected stopping time of their practical test, both of which we establish in Theorems 3.2 and 3.3. We believe that these differences—namely, the design of a different family of stochastic processes and the selection of $\lambda$—are essential for deriving our theoretical guarantees.

### 3.3 Practical DP auditing

While intuitive, the test in Section 3.2 requires oracle access to $\lambda^*$ and $f^*$. We now instantiate a practical algorithm inspired by the ideas developed in the previous subsection. Let's start assuming that we know $f^*$ but want to learn $\lambda^*$. If we construct the process

$$\mathcal{K}_0^* = 1, \quad \mathcal{K}_t^* = \mathcal{K}_{t-1}^* \times (1 + \lambda_t[f^*(X_t) - f^*(Y_t) - \tau(\varepsilon, \delta)]),$$

it is easy to see that by restricting the range of $\lambda_t$ as in the proof of Theorem 3.2, $\mathcal{K}_t^*$ is a nonnegative supermartingale so long as $\lambda_t$ is predictable (i.e, choosen before observing $X_t, Y_t$), implying that, with high probability, $\mathcal{K}_t^*$ remains bounded over time under $H_0$. Next, we need to choose a predictable sequence $\{\lambda_t\}_{t \geq 1}$ that ensures that $\mathcal{K}_t$ grows rapidly under $H_1$. This can be done by running ONS with losses $\ell_t^*(\lambda) = -\log(1 + \lambda[f^*(X_t) - f^*(Y_t) - \tau(\varepsilon, \delta)])$. The regret bound of ONS from [22], summarized in Theorem E.1, implies that for any $\lambda$,

$$\sum_{i \in [t]} \ell_i^*(\lambda) - \sum_{i \in [t]} \ell_i^*(\lambda_i) \leq O(\log(t)).$$

Noticing that $\sum_{i \in [t]} \ell_i^*(\lambda) = \log(\mathcal{K}_t^*(\lambda))$, where $\log(\mathcal{K}_t^*(\lambda))$ is the process from Equation (3) and $\sum_{i \in [t]} \ell_i^*(\lambda_i) = \log(\mathcal{K}_t^*)$, we have that

$$\lim_{t \to \infty} \frac{\log(\mathcal{K}_t^*)}{t} = \lim_{t \to \infty} \frac{\log(\mathcal{K}_t^*(\lambda^*))}{t} = \Omega(\Delta^2),$$

where the last equality comes from Theorem 3.2, Part 2. Hence, learning $\lambda^*$ with ONS does not hurt *asymptotically* if we know $f^*$. Finally, $f^*$ can be learned by running OGA with losses $h_t(f) = \langle f, K(X_t, \cdot) - K(Y_t, \cdot) \rangle_{\mathcal{H}}$:

$$f_1 = 0_{\mathcal{H}}, \quad f_{t+1} = \Pi(f_t + \eta_t \nabla h_t(f_t)),$$

where $\Pi$ is an operator projecting onto to the set of functions with $\|f\|_{\mathcal{H}} \leq 1$. Incorporating these changes, on the abstract test of the previous section, we arrive at the practical test presented in Algorithm 1, whose statistical properties are presented in Theorem 3.3.

---

**Algorithm 1** Sequential DP Auditing

---

1: **Input:** Neighboring datasets $S, S' \in \mathcal{D}$, mechanism $\mathcal{A}$, privacy parameters $\varepsilon, \delta$, maximum number of iterations $N_{\max}$.
2: Set $\tau(\varepsilon, \delta) = \sqrt{2}\left(1 - \frac{2(1-\delta)}{1+e^\varepsilon}\right)$
3: Initialize $\mathcal{K}_0 = 1, \lambda_1 = 0, f_1 = 0_{\mathcal{H}}$
4: **for** $t = 1, 2, ..., N_{\max}$ **do**
5:      Observe $X_t \sim \mathcal{A}(S), Y_t \sim \mathcal{A}(S')$
6:      $\mathcal{K}_t = \mathcal{K}_{t-1}\left(1 + \lambda_t\left[\langle f_t, K(X_t, \cdot) - K(Y_t, \cdot)\rangle_{\mathcal{H}} - \tau(\varepsilon, \delta)\right]\right)$
7:      **if** $\mathcal{K}_t \geq 1/\alpha$ **then**
8:          Reject $H_0$
9:      **else**
10:          Send $h_t(f_t) = \langle f_t, K(X_t, \cdot) - K(Y_t, \cdot)\rangle_{\mathcal{H}}$ to OGA, receive $f_{t+1}$
11:          Send $\ell_t(\lambda_t) = -\log(1 + \lambda_t[\langle f_t, K(X_t, \cdot) - K(Y_t, \cdot)\rangle_{\mathcal{H}} - \tau(\varepsilon, \delta)])$ to ONS, get $\lambda_{t+1}$
12:      **end if**
13: **end for**

---

**Theorem 3.3** (Statistical properties of Algorithm 1). *Suppose OGA in Line 10 is Algorithm 4 initialized on input $\{f \in \mathcal{H} : \|f\|_{\mathcal{H}} \leq 1\}, 0_{\mathcal{H}}$ and ONS in Line 11 is Algorithm 3 initialized on input $\left[0, \frac{1}{4+2\tau(\varepsilon, \delta)}\right]$. Let $\mathcal{K}_t$ be the process constructed in Algorithm 1 and $\mathcal{T} = \min\{t \geq 1 : \mathcal{K}_t \geq 1/\alpha\}$ be the stopping time of the test when $N_{max} = \infty$. Then,*

1. *Under $H_0$, $\mathbb{P}(\mathcal{T} < \infty) \leq \alpha$.*

2. *Under $H_1$, (i) $\lim_{t\to\infty} \frac{\log(\mathcal{K}_t)}{t} = \Omega(\Delta^2)$ and (ii) $\mathbb{E}[\mathcal{T}] = O\left(\frac{\log(1/\Delta)}{\Delta} + \frac{\log(1/(\alpha\Delta^2))}{\Delta^2}\right)$.*

Theorem 3.3 mimics the guarantees in Theorem 3.2 for the practical test in Algorithm 1. Item (1) states that under $H_0$, the probability of rejecting $H_0$ (which occurs if the stopping time $\mathcal{T}$ is finite) is upper-bounded by $\alpha$, thereby controlling the Type I error. Item (2) states that under $H_1$, the process $\mathcal{K}_t$ grows roughly as $\exp(t\Delta^2 - o(t))$ for sufficiently large $t$. This behavior is similar to that of the abstract test (Theorem 3.2), though practical learning aspects, such as the $o(t)$ term due to learning $\lambda^*$ with the Online Newton Step (ONS) algorithm, should be considered for the precise constants involved. Item (3) guarantees that the expected time to reject $H_0$, denoted $\mathbb{E}[\mathcal{T}]$ (i.e., the expected time to detect that an algorithm does not satisfy its privacy guarantee), is bounded by $1/\Delta^2$. Thus, the larger the MMD between the distributions generated by the mechanism on adjacent datasets (which contributes to a larger $\Delta$), the faster a privacy violation will be detected. Notably, our test does not need knowledge of $\Delta$, it adapts automatically.

## 3.4   An alternative approach based on e-processes

Algorithm 1 is a sequential test based on the construction of a nonnegative supermartingale and an application of Ville's inequality. E-processes are a class of stochastic processes broader than nonnegative supermatingales that can also be combined with Ville's inequality to design sequential tests. Concretely, an e-process is a nonnegative stochastic process almost surely dominated by a nonnegative supermartingale.

In concurrent work, [51] studied a general class of e-processes. An important sub-class are the ones of the form

$$W_t(\beta_1^t) = \prod_{i \in [t]} (1 + \beta_i(E_i - 1)) = \prod_{i \in [t]} \langle (\beta_i, 1 - \beta_i), (E_i, 1) \rangle, \tag{4}$$

where for all $i \geq 1$, $\beta_i \in [0, 1]$ and $E_i$ is an e-value for the null, meaning that $E_i$ is almost surely nonnegative and $\mathbb{E}_{H_0}[E_i] \leq 1$. If the coefficients $\{\beta_i\}_{i\geq 1}$ are predictable, then the process above is a nonnegative supermartingale under the null. We could employ ONS as in Algorithm 1 to approximate the fixed $\beta^*$ that makes $\log(W_t(\beta^*))$ grow the fastest under the alternative. However, this requires truncating the domain, since otherwise the gradients of $\log(1 + \beta(E_i - 1))$ with respect to $\beta$ explode when $1 + \beta(E_i - 1) \approx 0$, breaking the Lipschitzness property needed to obtain theoretical guarantees with ONS. [51] shows that choosing the sequence $\{\beta_i\}_{i\geq 1}$ according to the Universal Portfolio (UP) algorithm [9] allows to optimize over the whole interval $[0, 1]$ while at the same time achieving a regret bound with smaller constants than ONS. Since UP is difficult to implement in practice, they present an e-process $\tilde{W}_t$ that is upper bounded by the nonnegative supermartingale $W_t^{\mathrm{UP}}$ which we would obtain by optimizing $W_t(\beta_1^t)$ with UP.

For our concrete problem, we can prove that for every $t \geq 1$, $E_t := \frac{2 + f_t(X_t) - f_t(Y_t)}{2 + \tau}$ is an e-value for the null if $\{f_t\}_{t\geq 1}$ are predictable. This allows to re-write the processes $\mathcal{K}_t$ from the previous sections in the form of eq. (4), and noting that the optimization of $\{\lambda_i\}_{i\geq 1}$ is equivalent to the optimization of $\{\beta_i\}_{i\geq 1}$, we are able to design a slightly different sequential DP auditing procedure, Algorithm 2. The details of the algorithm, its statistical properties, and an experimental comparison with Algorithm 1 are provided in Appendix C.

# 4   Experiments

Below we provide empirical validation of the proposed sequential DP auditing framework in Algorithm 1.[2] We evaluate both Type I error control and detection power of this approach across different real-world DP mechanisms. First, we assess the performance of Algorithm 1 on both DP-compliant and non-DP mechanisms for computing the mean with additive noise. Moving forward, we demonstrate how DP-SGD implementations can be efficiently evaluated for privacy guarantees in under one complete training run. In Appendix F we provide additional empirical validation of

---

[2]The code to replicate our experiments is publicly available: https://github.com/google-research/google-research/tree/master/dp_sequential_test

the theoretical properties of our MMD-based tester using synthetic data with known underlying distributions, specifically Gaussian and perturbed uniform [42] distributions.

## 4.1 Additive-Noise Mechanisms for Mean Estimation

We evaluate our auditing methodology on mechanisms that employ additive noise when computing the mean. We analyze approaches using both Laplace and Gaussian noise distributions. Following the setup in [26], we define the following candidate DP mechanisms:

$$\text{DPLaplace}(X) := \frac{\sum_{i=1}^{n} X_i}{\tilde{n}} + \rho_1, \tag{5}$$

$$\text{NonDPLaplace1}(X) := \frac{\sum_{i=1}^{n} X_i}{n} + \rho_2, \tag{6}$$

$$\text{NonDPLaplace2}(X) := \frac{\sum_{i=1}^{n} X_i}{n} + \rho_1, \tag{7}$$

where $\tilde{n} = \max\{10^{-12}, n + \tau\}$, with $\tau \sim \text{Laplace}(0, 2/\varepsilon)$, $\rho_1 \sim \text{Laplace}(0, 2/[\tilde{n}\varepsilon])$, and $\rho_2 \sim \text{Laplace}(0, 2/[n\varepsilon])$. Among these mechanisms, only DPLaplace satisfies $\varepsilon$-DP. NonDPLaplace1 fails to maintain privacy guarantees by directly utilizing the private sample size $n$, while NonDPLaplace2 privatizes the number of samples when determining noise scale but calculates the mean using the non-privatized count. Additionally, we examine Gaussian noise variants—DPGaussian, NonDPGaussian1, and NonDPGaussian2—which correspond respectively to the analogous mechanisms but use additive Gaussian noise distributions.

We use the sequential DP tester to evaluate the proposed mechanisms for $\varepsilon \in \{0.01, 0.1\}$, controlling the additive noise introduced by each mechanism. For each setting, we test the null hypothesis that the mechanism satisfies $(\varepsilon, \delta)$-DP using the characterization in Definition 3.2 against the alternative that it does not. For this set of experiments, we fix the neighboring datasets to $S = \{0\}$ and $S' = \{0, 1\}$, although the sequential test remains agnostic of the specific choice of neighboring datasets. Moreover, we use 20 initial samples to set the bandwidth for the MMD tester using the median of the pairwise distances [17], which are then excluded from the actual testing phase to maintain statistical validity. We repeat each experiment 20 times and report the aggregated findings to ensure robust results and account for statistical variability. We report a failure to reject the null (no violation detected) when the test reaches 2,000 observations for $\varepsilon = 0.01$ and 5,000 samples for $\varepsilon = 0.1$.

Table 1 demonstrates the effectiveness and efficiency of the sequential DP auditing approach in identifying both compliant and non-compliant DP mechanisms across different privacy regimes. For both private mechanisms—DPGaussian and DPLaplace—we successfully control Type I error with zero rejections for both $\varepsilon = 0.01$ and $\varepsilon = 0.1$. In contrast, all non-DP mechanisms show significant rejection rates in high-privacy regimes ($\varepsilon = 0.01$), with DPGaussian1, NonDPLaplace1, and NonDPLaplace2 achieving 100% detection success using fewer than 350 observations on average, while NonDPGaussian2 reaches a 85% rejection rate requiring approximately 1,150 samples. This efficiency is particularly notable when compared to the fixed-sample-size MMD-tester recently proposed in [26], which failed to identify violations in the NonDPLaplace2 and NonDPGaussian2 mechanisms even when using 500,000 observations for $\varepsilon = 0.01$. The power of the sequential test decreases moderately for $\varepsilon = 0.1$, though it still maintains nearly perfect rejection rates for NonDPGaussian1, NonDPLaplace1 and NonDPLaplace2 mechanisms using fewer than 1,000 data points. Interestingly, NonDPGaussian2 presents the most challenging detection scenario, with rejection rates dropping to just 5% for $\varepsilon = 0.1$, suggesting its violations become more subtle as privacy constraints relax. We provide a detailed comparison with the fixed-sample-size approach from [26] in Appendix D.

## 4.2 DP-SGD Auditing in Less-Than-One Training Run

In this section, we demonstrate our sequential framework's ability to audit DP-SGD [1, 46] and identify privacy violations in less than one complete training run. We adopt a white-box auditing approach that grants access to intermediate gradients. This is more efficient than black-box audits, which require re-running the entire training process to generate each sample, making them computationally intensive.

Table 1: Algorithm 1 sequential DP auditing performance on mean mechanisms with additive Gaussian and Laplace noise across privacy regimes. Rejection rates indicate the proportion of experiments ($\pm$ standard errors over 20 independent runs) where algorithm 1 rejects the null hypothesis of $(\varepsilon, \delta)$-DP. $\bar{N}$ represents the average number of samples required to detect a violation, when it occurred ($\pm$ standard errors). Dashes (–) indicate that no violations were detected, consistent with true DP-mechanisms.

| | $\varepsilon = 0.01$ | | $\varepsilon = 0.1$ | |
|---|---|---|---|---|
| Mechanism | Rejection rate | $\bar{N}$ to reject | Rejection rate | $\bar{N}$ to reject |
| DPGaussian | $0.0 \pm 0.0$ | – | $0.0 \pm 0.0$ | – |
| NonDPGaussian1 | $1.0 \pm 0.0$ | $264 \pm 9.3$ | $1.0 \pm 0.0$ | $562 \pm 29.2$ |
| NonDPGaussian2 | $0.85 \pm 0.08$ | $1139 \pm 126.1$ | $0.05 \pm 0.05$ | $4776 \pm 219.0$ |
| DPLaplace | $0.0 \pm 0.0$ | – | $0.0 \pm 0.0$ | – |
| NonDPLaplace1 | $1.0 \pm 0.0$ | $331 \pm 14.5$ | $1.0 \pm 0.0$ | $920 \pm 61.6$ |
| NonDPLaplace2 | $1.0 \pm 0.0$ | $192 \pm 18.4$ | $0.95 \pm 0.05$ | $770 \pm 262.3$ |

The proposed sequential test not only detects violations but can also estimate a lower bound on the privacy parameter $\varepsilon$ by finding the smallest $\varepsilon$ that our the test fails to reject at a fixed $\delta$ and confidence level. Crucially, if the null hypothesis is never rejected, our method is equivalent to a standard one-run MMD audit on the full set of gradients, ensuring it is at least as accurate as comparable non-sequential methods while offering the potential for much faster detection.

**Auditing methodology.**   We adopt the white-box auditing procedure introduced as Algorithm 2 in [33]. At each training step $t$ a batch $B_t$ of examples is sampled. The auditor computes two private gradients: $\tilde{\nabla}[t]$ from the standard gradient $B_t$ and $\tilde{\nabla}'[t]$ where a canary gradient $g'$ was inserted with probability $q_c$. This canary is constructed to be orthogonal to all other per-example gradients within its batch in expectation. The auditor collects one-dimensional samples $x_t, y_t$, corresponding to the dot product between $g'$ and the gradients: $x_t = \langle g', \tilde{\nabla}[t] \rangle$ and $y_t = \langle g', \tilde{\nabla}'[t] \rangle$. Since the clipping norm and batch size are known by the auditor in the white-box setting, the samples are rescaled. This process yields $\{x_t\}_t$ drawn from a Gaussian distribution $P_1 = N(0, \sigma^2)$ and $\{y_t\}_t$ where the canary was present, drawn from $P_2 = N(1, \sigma^2)$. In practice, computing the canary described can be computationally intensive, so Dirac canary gradients—a gradient with zeros everywhere except at a single index—are used instead. This leads to samples $\{x_t\}_t, \{y_t\}_t$ that are not necessarily identically distributed over time. However, note that our algorithm can handle time-varying distributions: the supermartingale property under the null hypothesis is preserved so long as $x_t$ and $y_t$ are close in distribution at every time-step $t \geq 1$. See Section V.A in [44] for more details on time-varying distributions.

To find the privacy lower bound, we run parallel auditing processes for a set of candidate parameters $\{\varepsilon_i\}$. Each process tests a null hypothesis $H_0 : MMD(P_1, P_2) < \tau(\varepsilon_i, \delta)$. The empirical lower bound after $t$ steps, is the smallest candidate $\varepsilon_i$ that the test fails to reject at a confidence level $\alpha = 0.05$.

**Experimental Results.**   We audit a DP-SGD mechanism with a 0.1 batch sampling rate. The estimated per-step lower-bound, $\varepsilon_{canary}$, does not account for the composition or subsampling amplification of the final model's total privacy cost ($q_c$ defined above is set to $q_c = 1$ in our experiments).

Figure 1 shows the results from five independent runs. For the private implementation (Figure 1a), our test correctly fails to reject the null hypothesis after 500 observations in all 5 runs, as expected for this confidence level, and confirming the mechanism satisfies its expected privacy of $\varepsilon_{canary} = 0.01$ (corresponding to a total $\varepsilon \approx 0.03$).

In contrast, for the non-private implementation (Figure 1b), the audit successfully detects violations. It rejects the hypothesis $H_0 : \varepsilon_{canary} \leq 0.01$ in an average of just 60 observations and $H_0 : \varepsilon_{canary} = 0.1$ in 75 observations. After 250 observations, our method establishes a privacy lower bound of

$\varepsilon = 0.43$ for this non-private mechanism, and $\varepsilon = 0.59$ after 2,500 observations (in Appendix G, Figure 7).

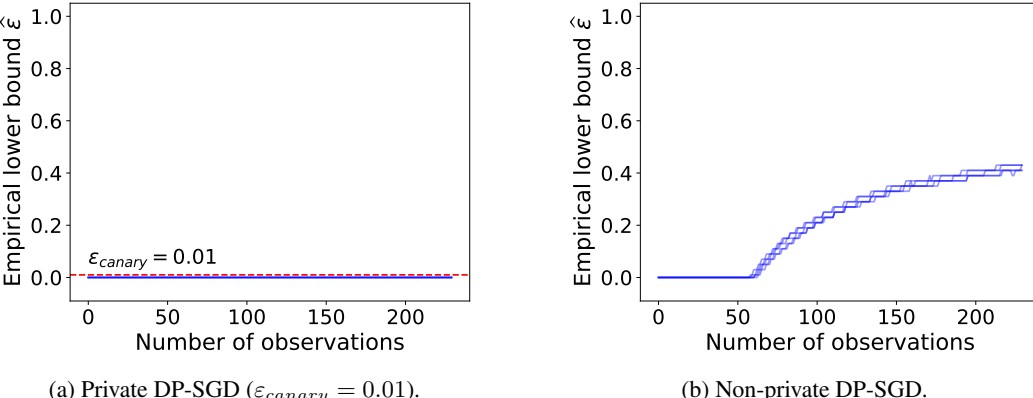

(a) Private DP-SGD ($\varepsilon_{canary} = 0.01$).        (b) Non-private DP-SGD.

Figure 1: Sequential audit results for DP-SGD implementations during training under white-box access with canary gradient threat model over 5 independent runs. Private implementations (left) are correctly identified as satisfying the specified differential privacy guarantee, while non-private implementations (right) are successfully detected as privacy violations for non-trivial values of $\varepsilon$.

**Discussion.** While effective, MMD-based tests face challenges when auditing large $\varepsilon$ values (e.g., $\varepsilon > 0.6$). In this regime, the MMD statistic and its rejection threshold $\tau$ both approach their theoretical maximum. This makes the gap, $\Delta^2$ (see Theorem 3.3), extremely small, leading to a large sample complexity, which is in the order of $1/\Delta^2$. We show in Appendix G (Figure 8) that when auditing a mechanism with a true $\varepsilon \approx 3$, the empirical lower bound grows very slowly, confirming this limitation.

Despite this, our experiments validate that our sequential test can monitor and identify privacy violations in DP-SGD dynamically during training. Unlike prior methods requiring at least one full training run, our approach provides empirical privacy bounds with only a few hundred gradient iterations. This represents a significant advancement in privacy auditing efficiency, enabling practitioners to verify privacy guarantees with substantially reduced computational cost.

## 5 Discussion

Our work introduces a sequential test for auditing differential privacy guarantees that automatically adapts to the unknown hardness of the testing problem stated in Definition 3.2. When compared with previous batch algorithms, our analysis and experiments show considerable improvements for detecting DP violations in non-private SGD and non-private mean estimation mechanisms. We also observe that auditors can efficiently characterize the privacy tradeoff function of a parameterized mechanism from a single observed stream by simultaneously testing multiple privacy levels —similar to techniques used for estimating means of bounded random variables via betting strategies [50]. We used this idea to compute empirical lower bounds on the privacy parameters in Section 4.

A first limitation of our approach, as observed for some DP-SGD audits, is the very large sample complexity required for large $\varepsilon$ privacy regimes. This is an inherent limitation of MMD-based auditing that also affects previous batch tests relying on this statistic. A second limitation, shared with prior work, is the reliance on a fixed pair of adjacent datasets. Ideally, we would like to combine our algorithm with a procedure that adaptively finds a worst-case pair.

Extending the sequential framework to audit larger privacy parameters or other important privacy definitions, such as Rényi DP or $f$-DP, represent valuable avenues for future investigation.

## Acknowledgments and Disclosure of Funding

We thank Jamie Hayes for useful guidance on auditing DP-SGD. We thank the anonymous reviewer who pointed out the connection between MMD and Total Variation, which helped us improve the MMD bound in Theorem 3.1. AR is supported by NSF grants DMS-2310718 and IIS-2229881. TG was partially funded to attend this conference by the CMU GSA/Provost Conference Funding.

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

# A Improvement on Lower-Bounding Hockey-Stick Divergence with the MMD

Theorem 4.13 in [26] introduces a lower bound on the MMD between two distributions that is parameterized by the order and magnitude of corresponding Hockey-Stick divergence. This connection divergence enables DP auditing with the MMD. Theorem 3.1 below tightens this bound . For completeness, we present first Theorem 4.13 in [26] and then a proof of Theorem 3.1.

**Lemma.** *([26, Theorem 4.13]) Let $\mathcal{H}$ be a reproducing kernel Hilbert space with domain $\mathcal{X}$ and kernel $K(\cdot, \cdot)$. Suppose $K(x, x) \leq 1$ for all $x \in \mathcal{X}$. If mechanism $\mathcal{A}$ is $(\varepsilon, \delta)$-DP, then for any $S \sim S'$*

$$\text{MMD}(\mathcal{A}(S), \mathcal{A}(S')) \leq e^\varepsilon - 1 + (e^{-\varepsilon} + 1)\delta. \tag{8}$$

**Theorem.** *(Theorem 3.1 in the main body). Let $\mathcal{H}$ be a reproducing kernel Hilbert space with domain $\mathcal{X}$ and kernel $K(\cdot, \cdot)$. Suppose $0 \leq K(x, y) \leq 1$ for all $x, y \in \mathcal{X}$. If mechanism $\mathcal{A}$ is $(\varepsilon, \delta)$-DP, then for any $S \sim S'$*

$$\text{MMD}(\mathcal{A}(S), \mathcal{A}(S')) \leq \sqrt{2} \left( 1 - \frac{2(1 - \delta)}{1 + e^\varepsilon} \right). \tag{9}$$

*Proof of Theorem 3.1.* Let $P = \mathcal{A}(S)$ and $Q = \mathcal{A}(S')$. It is well-known that

$$\text{MMD}(P, Q) = \|\mu_P - \mu_Q\|_{\mathcal{H}},$$

where $\mu_P = \mathbb{E}_{X \sim P}[K(\cdot, X)]$ and $\mu_Q = \mathbb{E}_{Y \sim P}[K(\cdot, Y)]$. It follows by Jensen's inequality that for any coupling $\pi$ between $P$ and $Q$

$$\begin{aligned}
\text{MMD}(P, Q) &= \|\mathbb{E}_{X \sim P}[K(\cdot, X)] - \mathbb{E}_{Y \sim Q}[K(\cdot, Y)]\|_{\mathcal{H}} \\
&= \|\mathbb{E}_{(X,Y) \sim \pi}[K(\cdot, X) - K(\cdot, Y)]\|_{\mathcal{H}} \\
&\leq \mathbb{E}_{(X,Y) \sim \pi}[\|K(\cdot, X) - K(\cdot, Y)\|_{\mathcal{H}}] \\
&\leq \left( \sup_{x,y \in \mathcal{X}} \|K(\cdot, x) - K(\cdot, y)\|_{\mathcal{H}} \right) \mathbb{E}_{(X,Y) \sim \pi}[\mathbb{1}_{X \neq Y}] \\
&= \left( \sup_{x,y \in \mathcal{X}} \|K(\cdot, x) - K(\cdot, y)\|_{\mathcal{H}} \right) \mathbb{P}_{(X,Y) \sim \pi}[X \neq Y].
\end{aligned}$$

First, we bound the term $\sup_{x,y \in \mathcal{X}} \|K(\cdot, x) - K(\cdot, y)\|_{\mathcal{H}}$. Note that for any $x, y \in \mathcal{X}$ we have

$$\begin{aligned}
\|K(\cdot, x) - K(\cdot, y)\|_{\mathcal{H}}^2 &= K(x, x) + K(y, y) - 2K(x, y) \\
&\leq 2(1 - K(x, y)) \\
&\leq 2,
\end{aligned}$$

where we have used that the assumption that $0 \leq K(x, y) \leq 1$ for all $x, y \in \mathcal{X}$. We conclude that $\sup_{x,y \in \mathcal{X}} \|K(\cdot, x) - K(\cdot, y)\|_{\mathcal{H}}^2 \leq 2$, which is equivalent to $\sup_{x,y \in \mathcal{X}} \|K(\cdot, x) - K(\cdot, y)\|_{\mathcal{H}} \leq \sqrt{2}$. Pluggin this into the MMD bound that we obtained above, we get that for any coupling $\pi$ between $P$ and $Q$

$$\text{MMD}(P, Q) \leq \sqrt{2} \mathbb{P}_{(X,Y) \sim \pi}[X \neq Y].$$

Minimizing the right hand side over all couplings, we obtain

$$\text{MMD}(P, Q) \leq \sqrt{2} \, \text{TV}(P, Q),$$

since $\inf_{\pi \in \Pi(P,Q)} \mathbb{P}_{(X,Y) \sim \pi}[X \neq Y] = \text{TV}(P, Q)$, where $\Pi(P, Q)$ denotes the set of all couplings between $P$ and $Q$ (see, e.g., [27, Proposition 4.7]). Finally, we use that

$$\text{TV}(P, Q) \leq 1 - \frac{2(1 - \delta)}{1 + e^\varepsilon},$$

as stated in [25, Remark A.1.b].

**Remark A.1.** *The* MMD *bound that we provide in Theorem 3.1 strictly improves over the one from [26, Theorem 4.13] for all privacy parameters $\varepsilon > 0, \delta \in (0,1)$:*

$$\sqrt{2}\left(1 - \frac{2(1-\delta)}{1+e^\varepsilon}\right) = \frac{\sqrt{2}}{1+e^\varepsilon}(1 + e^\varepsilon - 2 + \delta)$$
$$\leq e^\varepsilon - 1 + \delta$$
$$< e^\varepsilon - 1 + \delta + \delta e^{-\varepsilon}$$
$$= e^\varepsilon - 1 + \delta(1 + e^{-\varepsilon}).$$

$\square$

# B  Proofs in Section 3

**Theorem** (Theorem 3.2 in the main body). *Let $f^*$ be the witness function for $MMD(\mathcal{A}(S)||\mathcal{A}(S'))$ (as defined in definition 2.3). Given samples $\{X_t, Y_t\}_{t\geq 1} \overset{iid}{\sim} \mathcal{A}(S) \times \mathcal{A}(S')$ and a fixed hyperparameter $\lambda > 0$, define the stochastic process $\{\mathcal{K}_t^*(\lambda)\}_{t\geq 0}$ as follows:*

$$\mathcal{K}_0^* = 1$$
$$\mathcal{K}_t^*(\lambda) = \mathcal{K}_{t-1}^*(\lambda) \times (1 + \lambda[f^*(X_t) - f^*(Y_t) - \tau(\varepsilon, \delta)]),$$

*where $\tau(\varepsilon, \delta)$, defined in definition 3.2, represents the expected upper bound on MMD under the null hypothesis. Then it holds that:*

1. *Under the null hypothesis $(H_0 : \text{MMD}(\mathcal{A}(S), \mathcal{A}(S')) \leq \tau(\varepsilon, \delta))$, for any $\lambda \in [0, 1/(2+\tau(\varepsilon,\delta))]$ and any $\alpha \in (0,1)$, we have $\mathbb{P}[\sup_{t\geq 1} \mathcal{K}_t^*(\lambda) \geq 1/\alpha] \leq \alpha$.*

2. *Under the alternative $(H_1 : \text{MMD}(\mathcal{A}(S), \mathcal{A}(S')) > \tau(\varepsilon, \delta))$, there exists a $\lambda^* \in [0, \frac{1}{8+4\tau(\varepsilon,\delta)}]$ such that $\lim_{t\to\infty} \frac{\log \mathcal{K}_t^*(\lambda^*)}{t} = \Omega(\Delta^2)$ almost surely, where $\Delta = \text{MMD}(\mathcal{A}(S), \mathcal{A}(S')) - \tau(\varepsilon, \delta)$.*

*Proof.* We prove each item separately.

*Part 1:* To start, is easy to see that under $H_0$, $\mathcal{K}_t^*$ is a supermartingale for any $\lambda \geq 0$; letting $\mathcal{F}_t = \sigma(X_1, Y_1, ..., X_t, Y_t)$ be the $\sigma$-algebra generated by the samples observed up to time $t$,

$$\mathbb{E}[\mathcal{K}_t^*(\lambda) \mid \mathcal{F}_{t-1}] = \mathcal{K}_{t-1}^*(\lambda)(1 + \lambda[\mathbb{E}[f^*(X_t) - f^*(Y_t)] - \tau(\varepsilon, \delta)])$$
$$= \mathcal{K}_{t-1}^*(\lambda)(1 + \lambda[MMD(\mathcal{A}(S), \mathcal{A}(S')) - \tau(\varepsilon, \delta)]) \leq \mathcal{K}_{t-1}^*(\lambda),$$

since $\lambda[MMD(\mathcal{A}(S), \mathcal{A}(S')) - \tau(\varepsilon, \delta)] < 0$ under the null, and thus $(1 + \lambda[MMD(\mathcal{A}(S), \mathcal{A}(S')) - \tau(\varepsilon, \delta)]) < 1$.

Furthermore, by definition $f^*$ is bounded $|f^*(X_t)| \leq 1$, implying that

$$f^*(X_t) - f^*(Y_t) - \tau(\varepsilon, \delta) \in [-2 - \tau(\varepsilon, \delta), 2 - \tau(\varepsilon, \delta)].$$

Hence, $\mathcal{K}_t$ is nonnegative as long as $0 \leq \lambda \leq \frac{1}{2+\tau(\varepsilon,\delta)}$. Consequently, Ville's inequality (theorem 2.1) indicates that for any $\alpha \in (0,1), \lambda \in [0, \frac{1}{2+\tau(\varepsilon,\delta)}], \mathcal{K}_t(\lambda)$ remains bounded by $1/\alpha$ over time with probability at least $1 - \alpha$.

*Part 2:* For this part we need to prove that under $H_1 : \Delta = \text{MMD}(\mathcal{A}(S), \mathcal{A}(S')) - \tau(\varepsilon, \delta) > 0$, almost surely, there exists $\lambda^* \in [0, \frac{1}{8+4\tau(\varepsilon,\delta)}]$ such that for sufficiently large $t$, $\mathcal{K}_t(\lambda^*) \geq \exp(\Theta(t\Delta^2))$.

To see this, first note that for any $\lambda \in [0, \frac{1}{4\alpha+2\tau(\varepsilon,\delta)}]$ and any realization $\{x_t, y_t\}_{t\geq 1}$ of $\{X_t, Y_t\}_{t\geq 1}$, we can deterministically lower bound $\log(\mathcal{K}_t)$ as follows

$$
\begin{aligned}
\log(\mathcal{K}_t^*(\lambda)) &= \log\left(\mathcal{K}_0^* \prod_{i\in[t]}(1 + \lambda[f^*(x_i) - f^*(y_i) - \tau(\varepsilon,\delta)])\right) \\
&= 0 + \sum_{i\in[t]}\log(1 + \lambda[f^*(x_i) - f^*(y_i) - \tau(\varepsilon,\delta)]) \\
&\geq \sum_{i\in[t]}\lambda[f^*(x_i) - f^*(y_i) - \tau(\varepsilon,\delta)] - \sum_{i\in[t]}(\lambda[f^*(x_i) - f^*(y_i) - \tau(\varepsilon,\delta)])^2,
\end{aligned}
$$

where the first equality follows from the definition of $\mathcal{K}_t^*$ and the second one from the product rule for logarithms, and the last inequality uses the bound $\log(1+x) \geq x - x^2$ for $x \in (-1/2, 1/2)$—which can be verified finding the critical points of $g(x) = \log(1+x) - x - x^2$—combined with the fact that for $\lambda \in [0, \frac{1}{4+2\tau(\varepsilon,\delta)}]$, $\lambda[f^*(x_i) - f^*(y_i) - \tau(\varepsilon,\delta)] \in [-1/2, 1/2]$ for all $i \geq 1$.

Now, define

$$
r(t) = \frac{1}{t}\sum_{i\in[t]}[f^*(x_i) - f^*(y_i)] - \mathrm{MMD}(\mathcal{A}(S), \mathcal{A}(S')) \tag{10}
$$

and suppose that $r(t) \to 0$ as $t \to \infty$.

It follows that

$$
\lambda^* := \frac{\sum_{i\in[t]}[f^*(x_i) - f^*(y_i) - \tau(\varepsilon,\delta)]}{(8 + 4\tau(\varepsilon,\delta))\sum_{i\in[t]}[f^*(x_i) - f^*(y_i) - \tau(\varepsilon,\delta)] + 2\sum_{i\in[t]}[f^*(x_i) - f^*(y_i) - \tau(\varepsilon,\delta)]^2} \tag{11}
$$

satisfies $\lambda^* \in [0, \frac{1}{8+4\tau(\varepsilon,\delta)}]$ whenever $\sum_{i\in[t]}[f^*(x_i) - f^*(y_i) - \tau(\varepsilon,\delta)] \geq 0$ since it has the form $\frac{c}{(8+4\tau(\varepsilon,\delta))c+d}$, for $c, d > 0$.

But for any $t$,

$$
\frac{1}{t}\sum_{i\in[t]}[f^*(x_i) - f^*(y_i) - \tau(\varepsilon,\delta)] = r(t) + \mathrm{MMD}(\mathcal{A}(S), \mathcal{A}(S')) - \tau(\varepsilon,\delta) = r(t) + \Delta.
$$

Since we assumed $r(t) \to 0$, there exists $t_0$ such that for all $t \geq t_0$, $|r(t)| \leq \Delta/2$, implying that $\frac{1}{t}\sum_{i\in[t]}[f^*(x_i) - f^*(y_i) - \tau(\varepsilon,\delta)] = r(t) + \Delta \geq \Delta/2$.

We conclude that if $r(t) \to 0$ as $t \to \infty$, then for sufficiently large $t$,

$$
\frac{1}{t}\sum_{i\in[t]}[f^*(x_i) - f^*(y_i) - \tau(\varepsilon,\delta)] \geq \Delta/2,
$$

which in particular implies that $\sum_{i\in[t]}[f^*(x_i) - f^*(y_i) - \tau(\varepsilon,\delta)] > 0$, and hence $\lambda^* \in [0, \frac{1}{8+4\tau(\varepsilon,\delta)}]$. It follows for any $t \geq t_0$, plugging this value into the lower bound for $\mathcal{K}_t^*(\lambda^*)$ gives that $\log(\mathcal{K}_t^*(\lambda^*))$ can be lower bounded by

$$
\frac{\left(\sum_{i\in[t]}[f^*(x_i) - f^*(y_i) - \tau(\varepsilon,\delta)]\right)^2}{4\left(2\sum_{i\in[t]}[f^*(x_i) - f^*(y_i) - \tau(\varepsilon,\delta)]^2 + (8 + 4\tau(\varepsilon,\delta))\sum_{i\in[t]}[f^*(x_i) - f^*(y_i) - \tau(\varepsilon,\delta)]\right)}
$$

$$
\geq \frac{(t\Delta/2)^2}{4\left(2\sum_{i\in[t]}[f^*(x_i) - f^*(y_i) - \tau(\varepsilon,\delta)]^2 + (8 + 4\tau(\varepsilon,\delta))\sum_{i\in[t]}[f^*(x_i) - f^*(y_i) - \tau(\varepsilon,\delta)]\right)}
$$

$$
\geq \frac{(t\Delta/2)^2}{4(32t + 16t)} = \frac{t\Delta^2}{768} = \Theta(t\Delta^2). \tag{12}
$$

Our reasoning allows to conclude that

$$
\lim_{t\to\infty}\frac{\log(\mathcal{K}_t^*(\lambda^*))}{t} = \Omega(\Delta^2)
$$

whenever $r(t) \to 0$ as $t \to \infty$. Finally, the strong law of large numbers implies that the realizations of $\{X_t, Y_t\}_{t \geq 1}$ that do not satisfy $r(t) \to 0$ as $t \to \infty$, where $r(t)$ is defined in Equation (10), have a probability measure of 0, and hence our conclusion is valid almost surely. $\qquad\square$

**Theorem.** *(Theorem 3.3 in the main body) Suppose OGA in Line 10 is Algorithm 4 initialized on input $\{f \in \mathcal{H} : \|f\|_{\mathcal{H}} \leq 1\}, 0_{\mathcal{H}}$ and ONS in Line 11 is Algorithm 3 initialized on input $\left[0, \frac{1}{4+2\tau(\varepsilon,\delta)}\right]$. Let $\mathcal{K}_t$ be the process constructed in algorithm 1 and $\mathcal{T} = \min\{t \geq 1 : \mathcal{K}_t \geq 1/\alpha\}$ be the stopping time of the test when $N_{max} = \infty$. Then,*

1. *Under $H_0$, $\mathbb{P}(\mathcal{T} < \infty) \leq \alpha$.*

2. *Under $H_1$, (i) $\lim_{t \to \infty} \frac{\log(\mathcal{K}_t)}{t} = \Omega(\Delta^2)$ and (ii) $\mathbb{E}[\mathcal{T}] = O\left(\frac{\log(1/\Delta)}{\Delta} + \frac{\log(1/(\alpha\Delta^2))}{\Delta^2}\right)$.*

*Proof.* We prove each item separately.

*Part 1:* The proof follows directly from the arguments used in part 1 of theorem 3.2, with the difference that $\lambda_t$ is no longer fixed. To handle this, we simply use the fact that since $\lambda_t$ is the output of ONS after observing losses $\ell_1, ..., \ell_{t-1}$, which only depend on $\{X_i, Y_i\}_{i \in [t-1]}$, then $\lambda_t$ is $\mathcal{F}_{t-1}$-measurable.

*Part 2:*

(i) We proceed similarly to the proof of Theorem 3.2, but now $\lambda_t$ and $f_t$ vary with $t$. The process $\mathcal{K}_t$ constructed by running algorithm 1 can be written as

$$\mathcal{K}_0 = 1,$$
$$\mathcal{K}_t = \mathcal{K}_{t-1} \times (1 + \lambda_t[f_t(X_t) - f_t(Y_t) - \tau(\varepsilon,\delta)]).$$

For any $\lambda \geq 0$, define the process $\mathcal{K}_t(\lambda)$ as follows

$$\mathcal{K}_0(\lambda) = 1,$$
$$\mathcal{K}_t(\lambda) = \mathcal{K}_{t-1}(\lambda) \times (1 + \lambda[f_t(X_t) - f_t(Y_t) - \tau(\varepsilon,\delta)]). \tag{13}$$

This is the process that we would obtain by running algorithm 1 with a fixed $\lambda$, instead of computing $\lambda_1, \lambda_2, ..., \lambda_t$ via ONS. As before, we can prove that for any $\lambda \in [0, \frac{1}{4+2\tau(\varepsilon,\delta)}]$ and any realization $\{x_t, y_t\}_{t \geq 1}$ of $\{X_t, Y_t\}_{t \geq 1}$, $\log(\mathcal{K}_t(\lambda))$ can be deterministically lower bounded as follows:

$$\log(\mathcal{K}_t(\lambda)) = \log\left(\mathcal{K}_0 \prod_{i \in [t]} (1 + \lambda[f_i(x_i) - f_i(y_i) - \tau(\varepsilon,\delta)])\right)$$
$$= 0 + \sum_{i \in [t]} \log(1 + \lambda[f_i(x_i) - f_i(y_i) - \tau(\varepsilon,\delta)])$$
$$\geq \sum_{i \in [t]} \lambda[f_i(x_i) - f_i(y_i) - \tau(\varepsilon,\delta)] - \sum_{i \in [t]} (\lambda[f_i(x_i) - f_i(y_i) - \tau(\varepsilon,\delta)])^2.$$

Similarly to eq. (11), define $\lambda^*$ as

$$\lambda^* = \frac{\sum_{i \in [t]}[f_i(x_i) - f_i(y_i) - \tau(\varepsilon,\delta)]}{(8 + 4\tau(\varepsilon,\delta))\sum_{i \in [t]}[f_i(x_i) - f_i(y_i) - \tau(\varepsilon,\delta)] + 2\sum_{i \in [t]}[f_i(x_i) - f_i(y_i) - \tau(\varepsilon,\delta)]^2}. \tag{14}$$

It follows that $\lambda^* \in [0, \frac{1}{8+4\tau(\varepsilon,\delta)}]$ whenever $\sum_{i \in [t]}[f_i(x_i) - f_i(y_i) - \tau(\varepsilon,\delta)] \geq 0$.

Furthermore, by the regret guarantees of OGA (Theorem E.2),

$$\sup_{f:\|f\|_{\mathcal{H}} \leq 1} \sum_{i \in [t]} \langle f - f_i, K(x_i, \cdot) - K(y_i, \cdot) \rangle_{\mathcal{H}} \leq 6\sqrt{t},$$

since the functions $h_t(f) = \langle f, K(x_t, \cdot) - K(y_t, \cdot) \rangle_{\mathcal{H}}$ are 2-Lipschitz, since $K(x, x) \leq 1$ implies that $\|K(x_t, \cdot) - K(y_t, \cdot)\|_{\mathcal{H}} \leq 2$, and the diameter of the optimization domain

$\{f : \|f\|_{\mathcal{H}} \leq 1\}$ is $\sup_{f_1,f_2:\|f_1\|_{\mathcal{H}},\|f_2\|_{\mathcal{H}} \leq 1} \|f_1 - f_2\|_{\mathcal{H}} = 2$. Applying the reproducing property of $K$ implies, in particular, the following inequality for the witness function

$$\frac{1}{t} \sum_{i \in [t]} [f^*(x_i) - f^*(y_i) - \tau(\varepsilon, \delta)] - \frac{6}{\sqrt{t}} \leq \frac{1}{t} \sum_{i \in [t]} [f_i(x_i) - f_i(y_i) - \tau(\varepsilon, \delta)]. \quad (15)$$

As in Equation (10), let $r(t)$ be defined as

$$r(t) = \frac{1}{t} \sum_{i \in [t]} [f^*(x_i) - f^*(y_i)] - \mathrm{MMD}(\mathcal{A}(S), \mathcal{A}(S')),$$

so inequality (15) can be re-written as

$$r(t) + \Delta - \frac{6}{\sqrt{t}} \leq \frac{1}{t} \sum_{i \in [t]} [f_i(x_i) - f(y_i) - \tau(\varepsilon, \delta)]. \quad (16)$$

Assume $r(t) \to 0$ as $t \to \infty$. This also implies $r(t) - \frac{6}{\sqrt{t}} \to 0$ as $t \to \infty$. Consequently, for sufficiently large $t$ we obtain that $\frac{1}{t} \sum_{i \in [t]} [f_i(x_i) - f(y_i) - \tau(\varepsilon, \delta)] \geq \Delta/2$. We then lower bound the wealth for this $t$ by substituting the value of $\lambda^*$ previously defined eq. (14).

Analogously to the proof of Theorem 3.2 (see eq. (12)), we obtain

$$\log(\mathcal{K}_t(\lambda^*)) \geq \frac{t\Delta^2}{768}. \quad (17)$$

Finally, we note that since $\lambda_1, ..., \lambda_t$ are chosen by running ONS with losses $\ell_t(\lambda) = -\log(1 + \lambda[f_t(x_t) - f_t(y_t) - \tau(\varepsilon, \delta)])$, the regret bound of ONS (see theorem E.1) implies that for any $\lambda \in [0, 1/(4 + 2\tau(\varepsilon, \delta))]$

$$\sum_{i \in [t]} \ell_i(\lambda) - \sum_{i \in [t]} \ell_i(\lambda_i) \leq 10 \log(t), \quad (18)$$

since $\ell_t(\lambda)$ is 1-exp-concave, $4 + 2\tau(\varepsilon, \delta)$-Lipschitz:

$$|\ell_t'(\lambda)| = \left| \frac{f_t(x_t) - f_t(y_t) - \tau(\varepsilon, \delta)}{1 + \lambda[f_t(x_t) - f_t(y_t) - \tau(\varepsilon, \delta)]} \right| \leq \frac{2 + \tau(\varepsilon, \delta)}{1/2},$$

and the optimization domain is an interval of length $1/(4 + 2\tau(\varepsilon, \delta))$. Noticing that $\sum_{i \in [t]} \ell_i(\lambda) = \log(\mathcal{K}_t(\lambda))$ and $\sum_{i \in [t]} \ell_i(\lambda_i) = \log(\mathcal{K}_t)$, we have that, for sufficiently large $t$,

$$\frac{t\Delta^2}{768} - \log(\mathcal{K}_t) \leq \log(\mathcal{K}_t(\lambda^*)) - \log(\mathcal{K}_t) \leq 10 \log(t),$$

where the first inequality follows by subtracting $\log(\mathcal{K}_t)$ to both sides in equation (17).

This in turn implies that $\lim_{t \to \infty} \frac{\log(\mathcal{K}^t)}{t} = \Omega(\Delta^2)$.

Recall that we used the fact that $r(t)$ converges to 0 as $t \to \infty$, but this occurs almost surely by the Law of Large Numbers, so our conclusion also holds almost surely. This finishes the proof.

(ii) The main difference between the previous result and this one is that to bound $\mathbb{E}[\mathcal{T}]$ under $H_1$ we need to carefully quantify how quickly the term $r(t)$ from Equation (10) converges to 0.

In this proof, we consider $r(t) = \frac{1}{t} \sum_{i \in [t]} [f^*(X_i) - f^*(Y_i)] - \mathrm{MMD}(\mathcal{A}(S), \mathcal{A}(S'))$. That is, $r(t)$ is a random variable, as opposed to Equation (10), where it was defined for a specific realization of $\{(X_i, Y_i)\}_{i \geq 1}$. We also define $\tilde{r}(t) = r(t) - \frac{6}{\sqrt{t}}$. Using the tail-sum formula for expectation, we can expand the expectation as

$$\mathbb{E}[\mathcal{T}] = \sum_{t \geq 1} \mathbb{P}[\mathcal{T} \geq t] \leq \sum_{t \geq 1} \mathbb{P}[\mathcal{K}_t < 1/\alpha]$$

$$\leq \sum_{t \geq 1} \mathbb{P}[\mathcal{K}_t < 1/\alpha \text{ and } \tilde{r}(t) \leq \Delta/2] + \mathbb{P}[\Delta/2 \leq \tilde{r}(t)]$$

$$\leq \sum_{t \geq 1} \mathbb{P}[\mathcal{K}_t < 1/\alpha \text{ and } \tilde{r}(t) \leq \Delta/2] + \mathbb{P}[\Delta/2 \leq r(t)].$$

First, we bound the term $\mathbb{P}[\Delta/2 \leq r(t)]$. Bernstein inequality implies that

$$\mathbb{P}\left[r(t) \geq \max\left\{2\sigma\sqrt{\frac{2\log(t)}{t}}, \frac{16\log(t)}{3t}\right\}\right] \leq \frac{1}{t^2},$$

where $\sigma^2 = \mathbb{E}[f^*(X_1) - f^*(Y_1) - \mathrm{MMD}(\mathcal{A}(S), \mathcal{A}(S'))]^2$, since $r(t)$ is the average of the i.i.d terms $Z_1, ..., Z_t$, where $Z_i = f^*(X_i) - f^*(Y_i) - \mathrm{MMD}(\mathcal{A}(S), \mathcal{A}(S'))$ is centered ($\mathbb{E}[Z_i] = 0$) and bounded ($|Z_i| \leq 4$).

Let

$$t_0 = \min\left\{t \in \mathbb{N} : t \geq 3, \quad \max\left\{2\sigma\sqrt{\frac{2\log(t)}{t}}, \frac{16\log(t)}{3t}\right\} \leq \Delta/2\right\}.$$

Since $t_0 \geq 3$, then $\max\left\{2\sigma\sqrt{\frac{2\log(t)}{t}}, \frac{16\log(t)}{3t}\right\}$ is decresing in $t$ for all $t \geq t_0$, and by definition of $t_0$, $\max\left\{2\sigma\sqrt{\frac{2\log(t_0)}{t_0}}, \frac{16\log(t_0)}{3t_0}\right\} \leq \Delta/2$. This implies that for any $t \geq t_0$

$$\mathbb{P}[r(t) \geq \Delta/2] \leq \mathbb{P}\left[r(t) \geq \max\left\{2\sigma\sqrt{\frac{2\log(t)}{t}}, \frac{16\log(t)}{3t}\right\}\right] \leq \frac{1}{t^2}.$$

For $t \leq t_0$ we trivially upper bound $\mathbb{P}[r(t) \geq \Delta/2] \leq 1$. Combining these bounds, we obtain that

$$\sum_{t \geq 1} \mathbb{P}[\Delta/2 \leq r(t)] \leq t_0 + \sum_{t \geq t_0} 1/t^2 \leq t_0 + \pi^2/6.$$

Lemma 3 in [44] states that for any $a > 0$ and $b \in (0, 4]$,

$$\min\left\{n \in \mathbb{N} : \frac{\log(bn)}{n} \leq a\right\} \leq 1 + \max\left\{20, \frac{2\log(2/a)}{a}\right\}.$$

which implies the following inequality:

$$\min\left\{n \in \mathbb{N} : n \geq 3, \frac{\log(bn)}{n} \leq a\right\} \leq 1 + \max\left\{20, \frac{2\log(2/a)}{a}\right\}.$$

This is true if the condition $\frac{\log(bn)}{n} \leq a$ is met for $n < 3$, since the left side is $n = 3$ and the right-hand side is at least 21; in the case where the minimum is achieved at $n \geq 3$, the condition $n \geq 3$ can be removed and the inequality follows directly from the original lemma.

Hence, we obtain that $t_0 = O\left(\frac{\log(4/\Delta)}{\Delta} + \frac{\sigma^2 \log(4\sigma^2/\Delta^2)}{\Delta^2}\right)$. It follows that,

$$\sum_{t \geq 1} \mathbb{P}[\Delta/2 \leq r(t)] = O\left(\frac{\log(4/\Delta)}{\Delta} + \frac{\sigma^2 \log(4\sigma^2/\Delta^2)}{\Delta^2}\right).$$

Next, let's bound $\mathbb{P}[\mathcal{K}_t \leq 1/\alpha$ and $\tilde{r}(t) \leq \Delta/2]$. Equation (16) implies that under the event $\tilde{r}(t) \leq \Delta/2$, $\frac{1}{t}\sum_{i \in [t]}[f_i(X_i) - f(Y_i) - \tau(\varepsilon, \delta)] \geq \Delta/2$ and hence the lower bound for $\log(\mathcal{K}_t(\lambda^*)) \geq t\Delta^2/768$ from Equation (12) is valid for any realization of $\{(X_i, Y_i)_{i \geq 1}\}$ under this event, and so is the lower bound $\log(\mathcal{K}_t) \geq t\Delta^2/768 - 10\log(t)$, by the regret

guarantees of ONS. From these implications, it follows that

$$\sum_{t\geq 1} \mathbb{P}[\mathcal{K}_t < 1/\alpha \text{ and } \tilde{r}(t) \leq \Delta/2]$$

$$\leq \sum_{t\geq 1} \mathbb{P}\left[\mathcal{K}_t < 1/\alpha \text{ and } \frac{1}{t}\sum_{i\in[t]}[f_i(X_i) - f_i(Y_i) - \tau(\varepsilon,\delta)] \geq \Delta/2\right]$$

$$\leq \sum_{t\geq 1} \mathbb{P}\left[\mathcal{K}_t < 1/\alpha \text{ and } \log(\mathcal{K}_t(\lambda^*)) \geq t\Delta^2/768\right]$$

$$\leq \sum_{t\geq 1} \mathbb{P}\left[\mathcal{K}_t < 1/\alpha \text{ and } \mathcal{K}_t \geq \exp\left(t\Delta^2/768 - 10\log(t)\right)\right]$$

$$\leq \sum_{t\geq 1} \mathbb{P}\left[\exp\left(t\Delta^2/768 - 10\log(t)\right) < 1/\alpha\right]$$

$$= \sum_{t\geq 1} \mathbb{1}_{\left\{\exp\left(\frac{t\Delta^2}{768} - 10\log(t)\right) < 1/\alpha\right\}} = t_1,$$

where $t_1 = \min\{n \in \mathbb{N} : \exp\left(\frac{t\Delta^2}{768} - 10\log(t)\right) \geq 1/\alpha\}$, since $\frac{t\Delta^2}{768} - 10\log(t)$ is increasing in $t$. Finally, if $\log(t)/t \leq \frac{\Delta^2/2}{7680}$, then $\frac{t\Delta^2}{768} - 10\log(t) \geq \frac{t\Delta^2/2}{768}$. Lemma 3 from [44] implies that $\log(t)/t \leq \frac{\Delta^2/2}{7680}$ for $t = \Omega(\frac{\log(1/\Delta^2)}{\Delta^2})$. In addition, it is easy to see that $\frac{t\Delta^2/2}{768} \geq \log(1/\alpha)$ for $t = \Omega\left(\frac{\log(1/\alpha)}{\Delta^2}\right)$. We conclude that $t = \Omega\left(\frac{\log(1/\Delta^2) + \log(1/\alpha)}{\Delta^2}\right)$ suffices to obtain

$$\exp\left(\frac{t\Delta^2}{768} - 10\log(t)\right) \geq \exp\left(\frac{t\Delta^2/2}{768}\right) = \exp(\log(1/\alpha)) = 1/\alpha.$$

This implies that $\mathbb{1}_{\left\{\exp\left(\frac{t\Delta^2}{768} - 10\log(t)\right) < 1/\alpha\right\}} = 0$ for $t = \Omega\left(\frac{\log(1/\Delta^2) + \log(1/\alpha)}{\Delta^2}\right)$, thus $t_1 = O\left(\frac{\log(1/\Delta^2) + \log(1/\alpha)}{\Delta^2}\right)$. We conclude that

$$\mathbb{E}[\mathcal{T}] \leq O(t_0 + t_1) = O\left(\frac{\log(4/\Delta)}{\Delta} + \frac{\sigma^2\log(4\sigma^2/\Delta^2)}{\Delta^2} + \frac{\log(1/\Delta^2) + \log(1/\alpha)}{\Delta^2}\right).$$

This completes the proof. $\qquad\qquad\qquad\qquad\qquad\qquad\qquad\qquad\qquad\qquad\qquad\qquad\square$

## C Detailed Explanation of Alternative Sequential Test based on E-processes

### C.1 Derivation of the alternative test

As mentioned in Section 3.4, [51] studied a general class of e-processes of the form $W_t(\beta_1^t) = \prod_{i\in[t]}(1 + \beta_i(E_i - 1))$, described in eq. (4). Recall that for every $i \geq 1$, $\beta_i \in [0,1]$ and $E_i$ is an e-value for the null, meaning that $E_i$ is nonnegative and $\mathbb{E}_{H_0}[E_i] \leq 1$.

For our concrete problem, we can prove that $E_t := \frac{2 + f_t(X_t) - f_t(Y_t)}{2 + \tau}$ is an e-value for the null if $\{f_t\}_{t\geq 1}$ are predictable. Indeed, note that since $f_t(X_t) - f_t(Y_t) \geq -2$, then $E_t \geq 0$. Moreover, under the null

$$\mathbb{E}[E_t] = \mathbb{E}\left[\frac{2 + f_t(X_t) - f_t(Y_t)}{2 + \tau}\right]$$

$$= \frac{2 + \mathbb{E}\left[\mathbb{E}[f_t(X_t) - f_t(Y_t) \mid \{(X_i, Y_i)\}_{i\in[t-1]}]\right]}{2 + \tau}$$

$$\leq \frac{2 + \mathbb{E}[\sup_{f\in\mathcal{H}}\mathbb{E}[f(X_t) - f(Y_t) \mid \{(X_i, Y_i)\}_{i\in[t-1]}]]}{2 + \tau}$$

$$= \frac{2 + \sup_{f\in\mathcal{H}}\mathbb{E}[f(X_t) - f(Y_t)]}{2 + \tau}$$

$$= \frac{2 + \mathrm{MMD}}{2 + \tau} \leq \frac{2 + \tau}{2 + \tau} = 1.$$

Hence, the process $W_t^{\text{UP}}$ that results from sequentially choosing $\{\beta_t\}_{t\geq 0}$ with the Universal Portfolio (UP) algorithm [9] is a nonnegative supermartingale under the null. The regret guarantees of UP (Theorem E.3) give

$$\max_{\beta\in[0,1]} \log(W_t(\beta)) - \log(W_t^{\text{UP}}) \leq \log(t+1)/2 + \log(2),$$

where $W_t(\beta) = \prod_{i\in[t]}(1 + \beta(E_i - 1))$. From the equation above, it follows that

$$\tilde{W}_t = \exp\left(\max_{\beta\in[0,1]} \log(W_t(\beta)) - \log(t+1)/2 + \log(2)\right) \leq W_t^{\text{UP}}. \tag{19}$$

Hence, $\tilde{W}_t$ is an e-process, but not a supermartingale because it maximizes over $\beta$ after observing $(X_t, Y_t)$. The reason to consider $\tilde{W}_t$ instead of $W_t^{\text{UP}}$ is that running UP is computationally involved and $\tilde{W}_t$ is a reasonable lower bound that only requires solving a simple one-dimensional optimization problem in order to compute it. This reasoning gives rise to the following algorithm. Note that the witness function $f^*$ is still learned with OGA, as in Algorithm 1.

---

**Algorithm 2** Sequential DP Auditing with an E-process

1: **Input:** Neighboring datasets $S, S' \in \mathcal{D}$, mechanism $\mathcal{A}$, privacy parameters $\varepsilon, \delta$, maximum number of iterations $N_{\max}$.
2: Set $\tau(\varepsilon, \delta) = \sqrt{2}\left(1 - \frac{2(1-\delta)}{1+e^{\varepsilon}}\right)$
3: Initialize $W_0(\beta) = 1, f_1 = 0_{\mathcal{H}}$
4: **for** $t = 1, 2, ..., N_{\max}$ **do**
5:     Observe $X_t \sim \mathcal{A}(S), Y_t \sim \mathcal{A}(S')$
6:     $W_t(\beta) = W_{t-1}(\beta)\left(1 + \beta\left[\frac{2 + f_t(X_t) - f_t(Y_t)}{2 + \tau(\varepsilon, \delta)} - 1\right]\right)$
7:     $\tilde{W}_t = \exp\left(\max_{\beta\in[0,1]} \log(W_t(\beta)) - \log(t+1)/2 - \log(2)\right)$
8:     **if** $\tilde{W}_t \geq 1/\alpha$ **then**
9:         Reject $H_0$
10:    **else**
11:        Send $h_t(f) = \langle f, K(X_t, \cdot) - K(Y_t, \cdot)\rangle_{\mathcal{H}}$ to OGA, receive $f_{t+1}$
12:    **end if**
13: **end for**

---

### C.2   Statiscal properties of the alternative test

The theoretical guarantees of Algorithm 2 are similar to the ones stated about Algorithm 1 in Theorem 3.3, but we expect Algorithm 2 to perform better in practice due to the advantages of UP against ONS – in particular, the facts that UP allows to optimize over the interval $[0,1]$ instead of $[0, 1/2]$ and that its regret bound has smaller constants. See Appendix C.3 for experiments on the practical improvements that arise from using Algorithm 2 instead of Algorithm 1.

**Theorem C.1.** *Suppose OGA in Line 10 is Algorithm 4 initialized on input $\{f \in \mathcal{H} : \|f\|_{\mathcal{H}} \leq 1\}, 0_{\mathcal{H}}$. Let $\tilde{W}_t$ be the process constructed in Algorithm 2 and $\mathcal{T} = \min\{t \geq 1 : \tilde{W}_t \geq 1/\alpha\}$ be the stopping time of the test when $N_{max} = \infty$. Then,*

1. *Under $H_0$, $\mathbb{P}(\mathcal{T} < \infty) \leq \alpha$.*

2. *Under $H_1$, (i) $\lim_{t\to\infty} \frac{\log(\tilde{W}_t)}{t} = \Omega(\Delta^2)$ and (ii) $\mathbb{E}[\mathcal{T}] = O\left(\frac{\log(1/\Delta)}{\Delta} + \frac{\log(1/(\alpha\Delta^2))}{\Delta^2}\right)$, where $\Delta = \text{MMD}(\mathcal{A}(S), \mathcal{A}(S')) - \tau(\varepsilon, \delta)$.*

*Proof.* We prove each item separately. For simplicity we drop the dependence of $\tau$ in $(\varepsilon, \delta)$.

*Part 1.* From eq. (19) we obtain that under $H_0$

$$\mathbb{P}(\mathcal{T} < \infty) = \mathbb{P}(\exists t : \tilde{W}_t > 1/\alpha) \leq \mathbb{P}(\exists t : W_t^{\text{UP}} > 1/\alpha) \leq \alpha,$$

where the last inequality follows from the fact that $W_t^{\mathrm{UP}}$ is a nonnegative supermartingale under $H_0$ and Ville's inequality (Theorem 2.1).

*Part 2.* We formally prove that there is a one-to-one map between the processes $\mathcal{K}_t(\lambda_t)$ with $\lambda_i \in [0, 1/(2+\tau)]$ for all $i \geq 1$ and $W_t(\beta_1^t)$ with $\beta_i \in [0, 1]$ for all $i \geq 1$. This follows from choosing $\beta_i = (2+\tau)\lambda_i$ for all $i \geq 1$:

$$
\begin{aligned}
\mathcal{K}_t &= \prod_{i \in [t]} \big(1 + \lambda_i(f_t(X_t) - f_t(Y_t) - \tau)\big) \\
&= \prod_{i \in [t]} \left(1 + \lambda_i(2+\tau)\frac{f_t(X_t) - f_t(Y_t) - \tau}{2+\tau}\right) \\
&= \prod_{i \in [t]} \left(1 - \lambda_i(2+\tau) + \lambda_i(2+\tau)\left[\frac{f_t(X_t) - f_t(Y_t) - \tau}{2+\tau} - 1\right]\right) \\
&= \prod_{i \in [t]} \left(1 - \lambda_i(2+\tau) + \lambda_i(2+\tau)\left[\frac{2 + f_t(X_t) - f_t(Y_t)}{2+\tau}\right]\right) \\
&= \prod_{i \in [t]} \big(1 + \lambda_i(2+\tau)(E_i - 1)\big) \\
&= W_t(\beta_1^t).
\end{aligned}
$$

This equivalence allows us to reproduce exactly the same proof that we used for Theorem 3.3, where the only differences that arise from replacing ONS by UP are

- We are now able to optimize $\mathcal{K}_t$ over $\lambda_i \in [0, 1/(2+\tau)]$ for all $i \geq 1$. Recall that in order to use ONS we constrained to $\lambda_i \in [0, 1/(4+2\tau)]$ so that the losses $\ell_t$ from Algorithm 1 were Lipschitz.

- Even though we are optimizing over a larger domain, the regret guarantees that we obtain are better than those of ONS. Indeed, using ONS we obtained

$$
\max_{\lambda \in [0, 1/(4+2\tau)]} \log(\mathcal{K}_t(\lambda)) - \log(\mathcal{K}_t) \leq 10 \log(t)
$$

  in (18), while the process $\tilde{W}_t$ by construction satisfies

$$
\begin{aligned}
\max_{\lambda \in [0, 1/(2+\tau)]} \log(\mathcal{K}_t(\lambda)) - \log(\mathcal{K}_t) &= \max_{\beta \in [0,1]} \log(W_t(\beta)) - \log(\tilde{W}_t) \\
&= \log(t+1)/2 + \log(2).
\end{aligned}
$$

Hence, the process $\tilde{W}_t$ from Algorithm 2 has two advantages over the process $\mathcal{K}_t$ from Algorithm 1: it maximizes over a wider domain and has smaller regret. These two properties, combined with the fact that both algorithms learn the witness function with OGA, allow us to prove the same properties that we obtained for Algorithm 1 in Theorem 3.3, up to (slighlty better) constants.

$\qquad\square$

### C.3 Experiments and comparison to Algorithm 1

In the following, we replicate the experiments from Section 4.1 using our alternative sequential test based on e-processes. Table 2 presents the performance of Algorithm 2 when auditing additive-noise mechanisms for mean estimation. Our results show improved performance over Algorithm 1 in effectively and efficiently identifying both compliant and non-compliant DP mechanisms across the different privacy regimes studied. Specifically, we observe improved rejection rates for NonDPGaussian2 ($\varepsilon = 0.01$), and NonDPGaussian2, NonDPLaplace1 and NonDPLaplace2 ($\varepsilon = 0.1$), as well as a decrease in the average number of samples required to detect a violation for all the studied mechanisms, with the exception of NonDPGaussian2 when $\varepsilon = 0.1$.

We also apply this sequential test based on e-processes to audit the private and non-private implementations of DP-SGD from Section 4.2. Figure 2 shows the empirical performance of Algorithm 2

Table 2: Algorithm 2 sequential DP auditing based on e-processes performance on mean mechanisms with additive Gaussian and Laplace noise. Rejection rates indicate the proportion of experiments ($\pm$ standard errors over 20 independent runs) where Algorithm 2 rejects the null hypothesis of $(\varepsilon, \delta)$-DP. $\bar{N}$ represents the average number of samples required to detect a violation, when it occurred ($\pm$ standard errors). Dashes (–) indicate that no violations were detected, consistent with true DP-mechanisms. Values in bold represent improvements over the performance of Algorithm 1.

| | $\varepsilon = 0.01$ | | $\varepsilon = 0.1$ | |
|---|---|---|---|---|
| Mechanism | Rejection rate | $\bar{N}$ to reject | Rejection rate | $\bar{N}$ to reject |
| DPGaussian | $0.0 \pm 0.0$ | – | $0.0 \pm 0.0$ | – |
| NonDPGaussian1 | $1.0 \pm 0.0$ | $\mathbf{92} \pm 6.72$ | $1.0 \pm 0.0$ | $\mathbf{187} \pm 16.8$ |
| NonDPGaussian2 | $\mathbf{0.9} \pm 0.06$ | $\mathbf{728} \pm 139.8$ | $\mathbf{0.15} \pm 0.08$ | $4475 \pm 307.4$ |
| DPLaplace | $0.0 \pm 0.0$ | – | $0.0 \pm 0.0$ | – |
| NonDPLaplace1 | $1.0 \pm 0.0$ | $\mathbf{106} \pm 9.8$ | $\mathbf{1.0} \pm 0.0$ | $\mathbf{340} \pm 42.0$ |
| NonDPLaplace2 | $1.0 \pm 0.0$ | $\mathbf{54} \pm 4.9$ | $\mathbf{1.0} \pm 0.0$ | $\mathbf{253} \pm 119.8$ |

with faster detection rates in non-private regimes (Figure 2b), while successfully identifying private implementations (Figure 2a). In detail, Algorithm 2 rejects the hypothesis $H_0 : \varepsilon_{canary} = 0.1$ in just 18 observations on average over 5 runs, and establishes an empirical lower bound of $\varepsilon = 0.79$ using 250 observations in the white-box auditing regime detailed in Section 4.2.

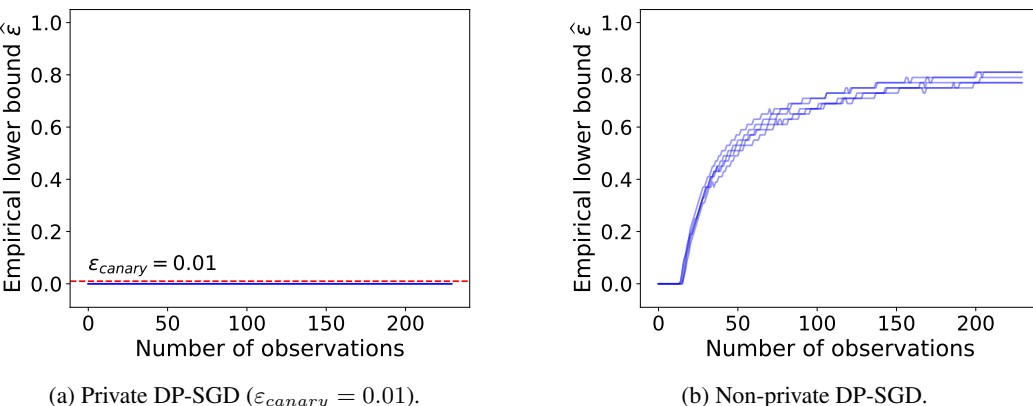

(a) Private DP-SGD ($\varepsilon_{canary} = 0.01$).    (b) Non-private DP-SGD.

Figure 2: Sequential audit results using e-process based testing (Algorithm 2) for DP-SGD implementations during training. White-box access with canary gradient threat model over 5 independent runs as in 4.2. The e-process approach demonstrates faster detection rates compared to Algorithm 1 while maintaining accurate identification of privacy-preserving mechanisms.

# D   Comparison to DP-Auditorium

In Section 4.1, we reported the performance of sequential tests at detecting privacy violations of algorithms for private mean estimation (Table 1) and commented on how these results compare to the ones reported in DP-Auditorium [26]. The presented sequential test differs from DP-Auditorium in many aspects, so in this section we decouple which of these aspects give the most improvement.

To make this comparison, we set $(\varepsilon, \delta) = (0.01, 10^{-5})$ and report the rejection rates in Table 3 over 10 runs of different auditing algorithms that gradually move from our sequential algorithm to the MMD-tester from [26]. Recall that the MMD-tester constructs a confidence interval $[L, U]$ for $\text{MMD}^2$ from a fixed batch of samples and then rejects if $L$ is greater than the square of the upper bound on the MMD. Below, we list key differences between this algorithms and our sequential procedure.

- Our algorithms use an improved bound on the MMD (see Theorem 3.1), which is tighter than the one used in [26].

- Both [26] and us use the RBF kernel. However, we use the median heuristic (MH) [17] to estimate the bandwidth using the first 20 samples, which are then not used in the auditing procedure, while in DP-Auditorium the authors set the bandwidth to a constant. For Table 3, we consider this constant to be 1.

- Our algorithms are sequential, which means that they can potentially detect privacy violations with smaller number of samples. Furthermore, we estimate the witness function using OGA while the MMD-tester just uses an empirical average.

In Table 3, we evaluate the following auditing procedures. The algorithm 'Batch' is the MMD-tester from [26]: it uses a fixed number of samples, the previous MMD bound from [26] and it fixes the bandwidth to 1. The algorithm 'Batch + new MMD bound' replaces the MMD bound by the one that we obtained in Theorem 3.1. The algorithm 'Batch + MH' modifies the MMD-tester by implementing the median heuristic. Finally, 'Batch + new MMD bound + MH' implements the MMD-tester along with the improved MMD bound and the median heuristic. Analogously, we define the sequential algorithms: 'Sequential' is an implementation of Algorithm 1 that uses the MMD bound from [26] and does not use the median heurstic, and so on.

From Table 3, we observe that some improvement comes from using a sequential algorithm[3], while some of it also comes from using the median heuristic. We note that the improvement on the MMD bound is not very significant, since for the privacy parameters that we chose both bounds are actually close; our bound is better when $\varepsilon$ is large (e.g., more than 1).

Table 3: We decouple the improvements of Algorithm 1 over the MMD-Tester from [26] by reporting detection rate of privacy violations with different auditing algorithms. We audit the privacy of three mechanisms defined in Section 4: a $(0.01, 10^{-5})$-DP mechanism (DPGaussian) and two non-DP mechanisms (NonDPGaussian1, NonDPGaussian2); we omit 'Gaussian' from their name in the table. We observe that some improvement comes from using our sequential algorithm, while some of it also comes from using the Median Heuristic (MH). We note that the improvement on the MMD bound is not very significant, since for the privacy parameters that we chose both bounds are actually close.

| Auditing Algorithm | DP | NonDP1 | NonDP2 |
|---|---|---|---|
| Sequential + new MMD bound + MH (our Algorithm 1) | 0 | 1 | 1 |
| Sequential + MH | 0 | 1 | 1 |
| Sequential + new MMD bound | 0 | 0 | 1 |
| Sequential | 0 | 0 | 1 |
| Batch + new MMD bound + MH | 0 | 1 | 0.1 |
| Batch + MH | 0 | 1 | 0.1 |
| Batch + new MMD bound | 0 | 0 | 0.5 |
| Batch (MMD-Tester [26]) | 0 | 0 | 0.5 |

# E  Technical Details

## E.1  Online Newton Step

Below, we present the Online Newton Step algorithm. In Algorithm 1, we run ONS with losses $\ell_t(\lambda) = -\log(1 + \lambda\left[\langle f_t, K(X_t, \cdot) - K(Y_t, \cdot)\rangle_{\mathcal{H}} - \tau(\varepsilon, \delta)\right])$.

---

[3]Note that this improvement is not necessarily achieved by every sequential test. For example, one could construct a confidence sequence $([L_t, U_t])$—a set of intervals such that with high probability, $L_t \leq \mathrm{MMD}^2 \leq U_t$ for all $t \geq 1$— and reject if $L_t$ is larger than the $\mathrm{MMD}^2$ upper bound [32]. This procedure would typically not improve on the batch test.

---

**Algorithm 3** Online Newton Step in 1D

---

1: **Input:** Interval $[a, b] \subseteq \mathbb{R}$
2: Set $\lambda_1 = 0, \beta = \frac{1}{2} \min\{1/(4L(b-a)), 1\}, A_0 = \frac{1}{\beta^2(b-a)^2}$
3: **for** $t = 1...$ **do**
4:      Play $\lambda_t$ and observe loss function $\ell_t$
5:      Compute gradient $g_t = \nabla \ell_t(\lambda_t)$
6:      Update scalar $A_t = A_{t-1} + g_t^2$
7:      $\lambda_{t+1} = \min\{b, \max\{a, \lambda_t - \frac{1}{\beta} \frac{g_t}{A_t}\}\}$
8: **end for**

---

**Theorem E.1** (Regret of ONS [22]). *Let $\lambda_1, \lambda_2, ...$ be the ONS iterates according to Algorithm 3. Suppose the functions $\ell_1(\cdot), \ell_2(\cdot), ...$ are L-Lipschitz and 1-exp-concave. Then, for every $t \geq 1$*

$$\max_{\lambda \in [a,b]} \sum_{i \in [t]} \ell_t(\lambda) - \ell_t(\lambda_t) \leq 10L(b-a)\log(t).$$

## E.2 Online Gradient Ascent

Below we present online gradient ascent. In Algorithm 1, we run OGA with losses $h_t(f) = \langle f, K(X_t, \cdot) - K(Y_t, \cdot) \rangle_{\mathcal{H}}$.

---

**Algorithm 4** Online Gradient Ascent in RHKS

---

  **Input:** Feasible set $\mathcal{G}, f_1 \in \mathcal{G}$
  **for** $t = 1, 2, \ldots$ **do**
     Play $f_t \in \mathcal{G}$, receive loss function $h_t$
     Compute gradient $g_t = \nabla h_t(f_t)$
     Update $f_{t+1} = \Pi_{\mathcal{G}}(f_t + \eta_t g_t)$
  **end for**

---

**Theorem E.2** (Regret of OGA [36]). *Denote $D = \sup_{f,g \in \mathcal{G}} \|f - g\|_{\mathcal{H}}$ and let $f_1, f_2, ...$ be the OGA iterates according to Algorithm 4. Suppose that the functions $h_1(\cdot), h_2(\cdot), ...$ are convex and L-Lipschitz, and $\eta_t = D/\sqrt{\sum_{i \in [t]} \|\nabla h_i(f_i)\|_{\mathcal{H}}^2}$ for every $t \geq 1$. Then, for every $t \geq 1$*

$$\max_{f \in \mathcal{G}} \sum_{i \in [t]} h_t(f) - h_t(f_t) \leq \frac{3D}{2} \sqrt{\sum_{i \in [t]} \|\nabla h_i(f_i)\|_{\mathcal{H}}^2} \leq \frac{3DL}{2} \sqrt{t}.$$

## E.3 OGA implementation

Note that OGA is not directly implementable. In particular the functions $f_t$ can't be stored in a computer. The only reason we need the functions $f_t$ in Algorithm 1 is to calculate $\langle f_t, K(X_t, \cdot) - K(Y_t, \cdot) \rangle_{\mathcal{H}} := v_t$. Below, we show how to do this. First, we find an expression for $f_t$. Note that $\Pi_{\mathcal{G}}(f) = \min\left\{\frac{1}{\|f\|_{\mathcal{H}}}, 1\right\} f$. Unrolling the recursion given by running OGA with losses $h_t(f) =$

$\langle f, K(X_t, \cdot) - K(Y_t, \cdot)\rangle_{\mathcal{H}}$ and denoting $M_t = \sum_{i \in [t]} \|K(X_i, \cdot) - K(Y_i, \cdot)\|_{\mathcal{H}}^2$ we obtain

$$f_{t+1} = \underbrace{\min \left\{ 1, \frac{1}{\left\| f_t + \frac{2[K(X_t, \cdot) - K(Y_t, \cdot)]}{\sqrt{M_t}} \right\|_{\mathcal{H}}} \right\}}_{:=\gamma_t} \left( f_t + \frac{2[K(X_t, \cdot) - K(Y_t, \cdot)]}{\sqrt{M_t}} \right)$$

$$= \gamma_t \left( f_t + \frac{2[K(X_t, \cdot) - K(Y_t, \cdot)]}{\sqrt{M_t}} \right)$$

$$= \gamma_t \left( \gamma_{t-1} \left( f_{t-1} + \frac{2[K(X_{t-1}, \cdot) - K(Y_{t-1}, \cdot)]}{\sqrt{M_{t-1}}} \right) + \frac{2[K(X_t, \cdot) - K(Y_t, \cdot)]}{\sqrt{M_t}} \right)$$

$$\vdots$$

$$= \sum_{i \in [t]} \left( \frac{2[K(X_i, \cdot) - K(Y_i, \cdot)]}{\sqrt{M_i}} \prod_{j=i}^{t} \gamma_j \right).$$

Hence, we obtain the following expression for $v_t$:

$$v_t = \langle f_t, K(X_t, \cdot) - K(Y_t, \cdot) \rangle_{\mathcal{H}} = 2 \sum_{i \in [t-1]} \left( \frac{K(X_i, X_t) - K(X_i, Y_t) - K(Y_i, X_t) + K(Y_i, Y_t)}{\sqrt{M_i}} \prod_{j=i}^{t-1} \gamma_i \right).$$

Since $M_t = \sum_{i \in [t]} \|K(X_i, \cdot) - K(Y_i, \cdot)\|_{\mathcal{H}}^2 = \sum_{i \in [t]} K(X_i, X_i)^2 - 2K(X_i, Y_i) + K(Y_i, Y_i)^2$ for every $t$, all the terms above are Kernel computations, except $\prod_{j=i}^{t} \gamma_i$. We find a computable expression for these terms.

Recall that $\gamma_t = \min \left\{ 1, \frac{1}{\left\| f_t + \frac{2[K(X_t, \cdot) - K(Y_t, \cdot)]}{\sqrt{M_t}} \right\|_{\mathcal{H}}} \right\}$, so we only need to compute the norm of the second term in the minimum. It follows that

$$\left\| f_t + \frac{2[K(X_t, \cdot) - K(Y_t, \cdot)]}{\sqrt{M_t}} \right\|_{\mathcal{H}}^2 = \|f_t\|_{\mathcal{H}}^2 + 4v_t/\sqrt{M_t} + 4(M_t - M_{t-1})/M_t$$

$$= \gamma_{t-1}^2 \left\| f_{t-1} + \frac{2[K(X_{t-1}, \cdot) - K(Y_{t-1}, \cdot)]}{\sqrt{M_{t-1}}} \right\|_{\mathcal{H}}^2 + 4v_t/\sqrt{M_t} + 4(M_t - M_{t-1})/M_t$$

$$\vdots$$

$$= 4v_t/\sqrt{M_t} + 4(M_t - M_{t-1})/M_t + 4 \sum_{i \in [t-1]} \left( \frac{v_i}{\sqrt{M_i}} + \frac{M_i - Mi - 1}{M_i} \right) \left( \prod_{j=i}^{t-1} \gamma_j \right)^2.$$

All of the terms $v_1, ..., v_t, M_1, ..., M_t, \gamma_1, ..., \gamma_{t-1}$ have already been computed by the time we need to compute $\gamma_t$. Hence, the above equality is a computable expression for $\gamma_t$.

### E.4 Universal Portfolio

The setup for the algorithm is the following. Consider two assets. At each time step $t = 1, 2, \ldots, T$, the market reveals a *price relative vector*

$$x_t = (x_{t,1}, x_{t,2}) \in \mathbb{R}_{>0}^2,$$

where $x_{t,i}$ is the ratio of the price of asset $i$ at time $t$ to its price at time $t-1$. A portfolio is a number $\beta \in [0, 1]$, where $\beta$ represents the fraction of wealth invested in asset 1 and $(1 - \beta)$ is invested in asset 2. The portfolio is *rebalanced* to the same allocation at each round.

The cumulative wealth achieved by a fixed portfolio $\beta \in [0, 1]$ after $t$ time steps is given by

$$W_t(\beta) := \prod_{s=1}^{t} (\beta x_{s,1} + (1 - \beta)x_{s,2}),$$

starting from initial wealth $W_0(\beta) = 1$. At time $t$, the universal portfolio strategy [9] defines a probability distribution over portfolios $\beta \in [0, 1]$, weighted by their wealth up to time $t - 1$. Specifically, under a prior density $\pi(\beta)$ on $[0, 1]$, the time-$t$ portfolio is chosen as

$$\beta_t := \frac{\int_0^1 \beta \cdot W_{t-1}(\beta)\,\pi(\beta)\,d\beta}{\int_0^1 W_{t-1}(\beta)\,\pi(\beta)\,d\beta}.$$

In our case, $\pi(\beta)$ is the **Beta**$(1/2, 1/2)$ density:

$$\pi(\beta) = \frac{1}{\pi} \cdot \frac{1}{\sqrt{\beta(1-\beta)}} \quad \text{for } \beta \in (0, 1).$$

The optimal constant-rebalanced portfolio is

$$\beta^* \in \arg\max_{\beta \in [0,1]} W_T(\beta).$$

The cumulative wealth of the universal portfolio strategy is

$$W_T^{\mathrm{UP}} := \prod_{t=1}^{T} (\beta_t x_{t,1} + (1 - \beta_t) x_{t,2}).$$

**Theorem E.3** (Regret of UP (Theorem 3 in [10])). *Let* $\beta^* \in [0, 1]$ *be the best fixed constant-rebalanced portfolio in hindsight:*

$$\beta^* \in \arg\max_{\beta \in [0,1]} W_t(\beta).$$

*Then the universal portfolio satisfies the regret bound:*

$$\log W_t(\beta^*) - \log W_t^{UP} \leq \frac{1}{2}\log(t + 1) + \log 2.$$

*This holds deterministically for any sequence of market vectors* $x_1, \ldots, x_t \in \mathbb{R}_{>0}^2$.

# F    Analysis of Algorithm 1 on Synthetic Data with Known Distributions

All the experiments presented in the main text and in the following subsections were conducted using Google Colab's standard CPU runtime environment (12.7 GB RAM) with Python 3.

## F.1    Perturbed uniform distributions

We begin by empirically demonstrating the behavior of our sequential test under two scenarios: when the null hypothesis of equal distributions is true and when it is false. For the former, we compare two 2-dimensional uniform distributions on the unit cube $[0, 1]^2$ where the auditing process $\mathcal{K}_t$ from Algorithm 1 remains bounded below the rejection threshold $1/\alpha$ for a statistical confidence level $\alpha = 0.05$. Figure 3 confirms that, as expected, we successfully control the Type I error at the desired statistical level when $H_0$ holds, across 100 simulations. For the latter scenario, we apply our test to compare a uniform distribution on $[0, 1]^2$ against a perturbed uniform distribution following the construction in [42] with one perturbation, across 100 simulations. In line with our theoretical predictions, the auditing process $\mathcal{K}_t$, which measures the cumulative evidence against the null hypothesis, grows exponentially under this alternative hypothesis, as illustrated in Figure 4. The test successfully rejects $H_0$ after, on average, collecting only 108 observations.

## F.2    Gaussian distributions

We then test our MMD-based sequential approach by comparing two Gaussian distributions while varying the dimensionality and separation between them. We fix one distribution as a $d$-dimensional standard normal distribution $\mathbb{P}_1 \sim \mathcal{N}(0, I_d)$, where $I_d$ is the $d$-dimensional identity matrix, varying $d$ from 1 to 5. Our test evaluates $H_0 : \mathrm{MMD}(\mathbb{P}_1, \mathbb{P}_2) = 0$ against $H_1 : \mathrm{MMD}(\mathbb{P}_1, \mathbb{P}_2) > 0$, with $\mathbb{P}_2 \sim \mathcal{N}(\mu, I_d)$ where we vary the center $\mu$ of the contrast distribution. We set the norm $||\mu||$ to different values between 0 to 1 to illustrate scenarios where the two distributions have different

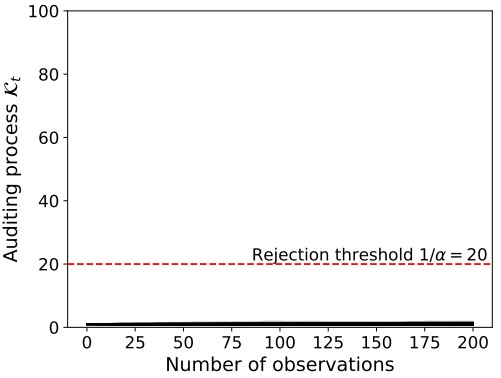
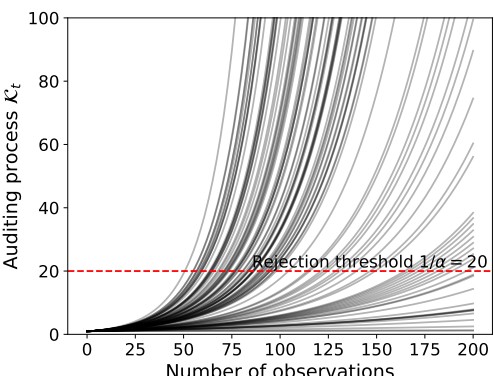

Figure 3: Type I error control in sequential testing. The auditing process for 100 simulations comparing two identical 2-dimensional uniform distributions on $[0,1]^2$. All trajectories remain below the rejection threshold (horizontal dashed line at $1/\alpha$), confirming proper Type I error control at the $\alpha = 0.05$ significance level.

Figure 4: Detection power under alternative hypothesis. We calculate the auditing process for 100 simulations comparing a uniform distribution on $[0,1]^2$ against a perturbed uniform distribution. The exponential growth of the auditing process leads to rejection of the null hypothesis after 108 observations on average.

degrees of separation, making it progressively harder for the test to find evidence against the null as $||\mu||$ is closer to 0. Note that when $||\mu|| = 0$, the null hypothesis is true.

Figure 5 shows the proportion of tests that reject the null hypothesis across these varying scenarios, based on 20 simulations for each setting. For a separation level of $||\mu|| \geq 0.5$, the test consistently rejects $H_0$ correctly, even in high-dimensional settings ($d = 5$). With minimal separation ($||\mu|| = 0.25$), the rejection rate is 62% when $d = 1$, degrading its power to 10% in higher dimensions. We also successfully control the Type I error when $H_0$ holds ($||\mu|| = 0$), rejecting in fewer than $\alpha = 0.05$ of the tests. Figure 6 illustrates the average number of samples needed to reject, further demonstrating the advantages of our proposed approach. Our sequential test correctly rejects $H_0$ with fewer than $\sim$700 observations, even in high dimensions, when the Gaussian distributions are well-separated ($||\mu|| \geq 0.5$), and with fewer than 1000 observations in low-separation settings.

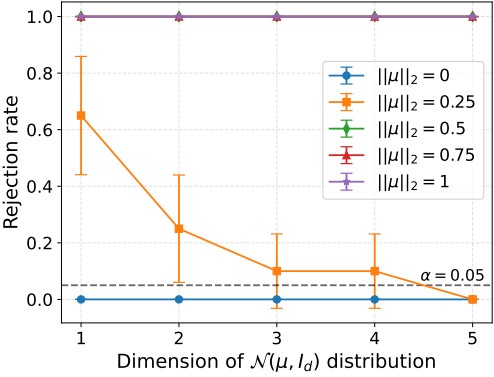
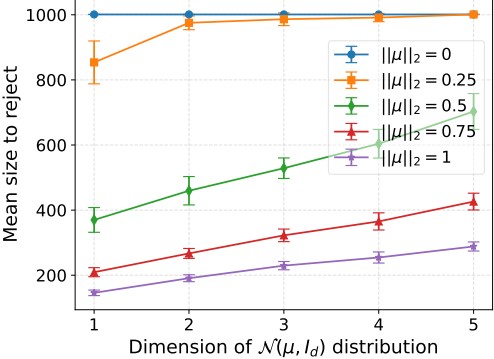

Figure 5: Power analysis of sequential MMD test across dimensions and separation levels. The plot shows rejection rates over 20 simulations comparing $\mathcal{N}(0, I_d)$ against $\mathcal{N}(\mu, I_d)$ with varying dimensionality $d$ and mean separation $\mu$. Type I error is controlled when $||\mu|| = 0$.

Figure 6: Sample efficiency of sequential MMD test. Average number of observations required for test rejection across different dimensionality ($d$) and separation ($\mu$) settings. Lower sample sizes demonstrate the test's efficiency at detecting distributional differences.

# G  Auditing DP-SGD for Certain Privacy Regimes

Below we provide additional empirical analysis of our sequential privacy auditing framework applied to DP-SGD implementations across different privacy regimes. We examine two key scenarios that complement our main results. First, Figure 7 audits non-private DP-SGD mechanisms over extended observation periods to demonstrate the evolution of our privacy lower bound estimates. Second, Figure 8 study the inherent limitations of MMD-based approaches when auditing mechanisms with large privacy parameters ($\varepsilon \approx 3$). These experiments illustrate both the strengths and practical limitations of our sequential testing methodology.

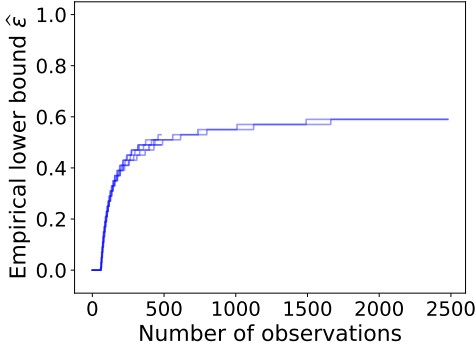

Figure 7: Extended sequential audit results of non-private DP-SGD implementations over 2,500 observations. The audit successfully rejects privacy hypotheses and establishes increasingly tight lower bounds on the privacy parameter.

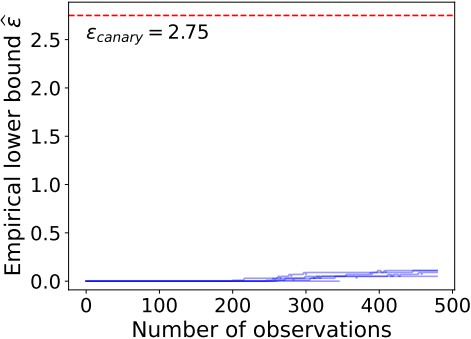

Figure 8: Sequential audit results for private DP-SGD implementations with large privacy parameters ($\varepsilon_{canary} = 2.75$, $\delta = 10^{-5}$). The empirical lower bound grows slowly due to the small gap $\Delta^2$ between the MMD and the rejection threshold $\tau(\varepsilon, \delta)$, demonstrating the sample complexity challenges in this regime.

# H  Detailed Comparison with Related Work

Verifying that a randomized mechanism satisfies a DP guarantee involves checking that a specific divergence (related to the privacy parameters $\varepsilon$ and $\delta$) remains bounded over all possible pairs of neighboring datasets. This task is often divided into two sub-problems: (1) identifying a "worst-case" or dominating pair of neighboring datasets, that maximizes the privacy loss, and (2) accurately estimating or bounding the divergence induced by this dominating pair. While our work focuses on the second sub-problem, we start by briefly review approaches for tackling the former. Although the concept of sequential privacy auditing appears new to our knowledge, we then provide a review of different methodologies within the field that are relevant to the second problem of divergence estimation and bounding. We finish by summarizing relevant work in the broader area of sequential hypothesis testing.

Identifying worst-case datasets has been tackled through methods like grid search over datasets [11, 5], explicit construction under strong parametric assumptions on the mechanism [14], black-box optimization techniques that iteratively search for high-privacy-loss inputs [26], or the use of predefined canaries [33]. More recent approaches showed efficient testing relying on randomized canaries for specific mechanisms like the Gaussian mechanism in high dimensions [2]. Black-box optimization approaches treat divergence maximization as an objective but often lack formal guarantees on finding the true worst case [26]. Our proposed auditing method can be paired with any technique for identifying candidate worst-case dataset pairs.

Given a candidate pair of neighboring datasets, the core challenge is to estimate the resulting privacy loss or divergence. There is substantial prior work on this, primarily situated within the batch setting, where a fixed number of samples are drawn a priori.

Estimating distances or divergences between distributions is a classical statistics problem [34, 47, 6]. Some methods employ optimization over function spaces, such as neural networks, to estimate $f$-divergences or Rényi divergences. While powerful, the finite-sample guarantees provided by some of these methods can depend on estimator properties (e.g., neural network architecture) and may become vacuous for DP auditing purposes [6]. Furthermore, many provide only asymptotic guarantees, whereas DP requires strict finite-sample bounds. Our work, in contrast, leverages sequential analysis to provide anytime-valid results with potentially much lower sample complexity.

Our approach relies on sequential testing with kernel methods, specifically the Maximum Mean Discrepancy (MMD)[19, 44], to construct test statistics. Prior work on privacy auditing has also used kernel methods, such as estimating regularized kernel Rényi divergence [13], but often requires strong assumptions (e.g., knowledge of covariance matrices) impractical in black-box settings or for mechanisms beyond Gaussian or Laplace.

Some auditing techniques require strong assumptions, like a discrete output space [18] or access to the mechanism's internal randomness or probability density/mass functions [12]. Others need access to the cdf of the privacy loss random variable [14]. StatDP [11] requires semi-black-box access (e.g., running the mechanism without noise) [11]. While our method can incorporate white-box information to improve power, it fundamentally operates in the black-box setting. The applicability of these methods that require strong assumptions is limited in the blackbox setting.

For the general black-box setting, tools like DP-Sniper [5], DP-Opt [35], Delta-Siege [28], and Eureka [29] perform black-box testing by searching for an output event maximizing the probability difference between neighboring inputs. However, DP-Sniper is specific to pure $\varepsilon$-DP, Delta-Siege for "$\rho-$ordered mechanisms" and all suffer from high sample complexity (millions of samples) in the batch setting.

A significant body of work focuses on auditing machine learning models, particularly those trained with DP-SGD. This includes membership inference attacks (MIA) [24, 37, 7, 23] and data reconstruction attacks [21, 4]. These methods often require white-box access to the model and sometimes large portions of the training data, but more importantly, work in the batch setting, requiring significant compute power. In the best case, they require training one full model end-to-end with canaries inserted every other step. For some easy to detect failures this might be wasting resources, and when the test fails, one has to start from scratch and previous samples cannot be reused for significant testing.

