# OpenReview forum: "Sequentially Auditing Differential Privacy"
_NeurIPS.cc/2025/Conference — NeurIPS 2025 poster_

### Official Review · Reviewer_9SwZ · 2025-06-24

**Clarity:** 4
**Significance:** 4
**Originality:** 4
**Rating:** 5
**Confidence:** 4

**Summary:**

Recently there has been a great deal of interest in "auditing" differential privacy guarantees of algorithms such as DP-SGD among others. The popular approach to auditing DP usually looks at sample size as static -- it runs the DP mechanism as a black-box n times with different randomness, and then provides an answer about its privacy properties. Usually, it provably accepts every DP mechanism, and empirically rejects non-private mechanisms with a certain amount of probability.

This work proposes a new framework and mechanism to audit DP mechanisms in a different, sequential way. If I understand correctly, here the number t of runs of the black-box is varying (and growing), and the algorithm either decides to terminate with a decision, or to keep asking for more samples. These tests are enabled through drawing a novel connection between differential privacy, and recent advances in sequential hypothesis testing. There is also a nice theorem on the relationship between the Hockey-stick divergence and kernel-MMD tests that is of independent interest.

**Questions:**

None.

**Ethical Concerns:**

["NO or VERY MINOR ethics concerns only"]

**Final Justification:**

This is a strong paper that combines novel ideas from disparate communities to prove a nice result. I strongly recommend acceptance!

**Quality:**

4

**Strengths And Weaknesses:**

Strengths:

+ This is a very strong and novel paper. The result is strong and elegant and arises due to a novel connections between two disparate areas. I also really liked the last part about auditing DPSGD in less than one run, which is a novel advance.
+ The paper is quite well-written, and even though the math is fairly dense it feels easy to read. Related works are well-cited and credit properly assigned.

Weaknesses:

- One weakness of the presentation is that this paper audits DP in a framework that is somewhat different from the standard auditing framework. It took me a while to understand this -- and I think the presentation could be even better if this were directly explained earlier on in the paper.
- A minor point -- the first DPSGD paper was actually Song, S., Chaudhuri, K., & Sarwate, A. D. (2013, December). Stochastic gradient descent with differentially private updates. In 2013 IEEE global conference on signal and information processing (pp. 245-248). IEEE.

---

> ### Author Rebuttal · Authors · 2025-07-31
>
> Thank you for the thoughtful review and helpful feedback. Below we address weaknesses and questions.
>
> # Weaknesses
>
> >`One weakness of the presentation is that this paper audits DP in a framework that is somewhat different from the standard auditing framework. It took me a while to understand this -- and I think the presentation could be even better if this were directly explained earlier on in the paper.`
>
> The hypothesis tested in the proposed sequential auditing framework coincides with previous work [A, B]. However, we take a different approach to testing this hypothesis: instead of fixing the sample size a priori  to construct a single confidence interval (or test), we allow $n$ to grow and run a sequential test that can be continuously monitored and adaptively stopped. We will explain this difference with prior work in more detail in the introduction.
>
> > `A minor point -- the first DPSGD paper was actually Song, S., Chaudhuri, K., & Sarwate, A. D. (2013, December). Stochastic gradient descent with differentially private updates. In 2013 IEEE global conference on signal and information processing (pp. 245-248). IEEE.`
>
> Thanks for pointing out the correct first DP-SGD paper. We will update the citation.
>
> [A] Kong, William, et al. "DP-Auditorium: A Large-Scale Library for Auditing Differential Privacy." 2024 IEEE Symposium on Security and Privacy (SP). IEEE, 2024.
>
> [B] Nasr, Milad, et al. "Tight auditing of differentially private machine learning." USENIX Security Symposium. 2023.

---

> > ### Comment · Reviewer_9SwZ · 2025-08-05
> > **thanks for the response**
> >
> > Thank you for the detailed response. i will keep my positive score.

---

### Official Review · Reviewer_codM · 2025-07-02

**Clarity:** 3
**Significance:** 2
**Originality:** 3
**Rating:** 4
**Confidence:** 3

**Summary:**

Given two neighbouring datasets $S$ and $S’$, a mechanism $\mathcal{A}$, and i.i.d samples $X_1, X_2, \dots \sim \mathcal{A}(S)$ and $Y_1, Y_2 \dots \sim \mathcal{A}(S’)$, this paper studies sequential hypothesis tests for distinguishing between:
- $H_0$: the Hockey-stick divergence of order $e^\epsilon$ between $\mathcal{A}(S)$ and $\mathcal{A}(S’)$ is smaller than $\delta$

V.S.

- $H_1$: the Hockey-stick divergence is $> \delta$.

First, to solve this, the paper uses the Maximum Mean Discrepancy (MMD) as a proxy for the Hockey-stick divergence. While this proxy was explored in prior work (e.g., [23]), the authors tighten the connection in Theorem 3.1. Then, the paper adapts the “two-sided” sequential MMD Test from [40] into a “one-sided” test against a specific non-zero threshold.  A practical sequential test is developed (Algorithm 1) and analysed with guarantees on Type I error, expected stopping time, and exponential growth under the alternative in Theorem 3.3.  Finally, the test is implemented to audit first some classic mean estimation DP mechanisms, then to audit DP-SGD in the white-box setting.

**Questions:**

- What exactly are the “non-trivial adaptations and subtle but key modifications of the original results
presented in [ 40 ]” (Line 140)? Could you explicitly enumerate or summarise what changes were needed to adapt the two-sided test to your one-sided setting?


- Improving further Theorem 3.1 via TV: In the proof of Theorem 3.1, it seems that you first upper bound the MMD by a Total Variation (TV). But it seems that Remark A.1.b) in the paper by Kairouz et al. (“The Composition Theorem for Differential Privacy”) provides tighter bounds on TV in terms of DP parameters. Can't you directly plug in those to yield an even tighter bound than your Theorem 3.1?

- Isn’t auditing DP-SGD in the white-box setting equivalent to auditing the mean mechanism applied to gradients, and thus section 4.2 is the same as section 4.1? If not, what are the differences?

- Is it possible to use your sequential test for auditing DP-SGD in the black-box model, i.e. you only have query access to the learned model and not access to gradients anymore?

**Ethical Concerns:**

["NO or VERY MINOR ethics concerns only"]

**Final Justification:**

The authors' rebuttal addressed my questions. I kept a borderline accept recommendation due to the assumption that the two neighbouring datasets are given as input to the auditing algorithm, rather than generated. This is the main limitation for this line of work, as generating worst-case neighbouring datasets is arguably the hardest part of auditing DP.

**Limitations:**

There is no specific "limiations" sections in the paper, but the authors discuss them throughout the paper

**Quality:**

2

**Strengths And Weaknesses:**

Strength:
- The paper is well-written and the problem is well-motivated
- The idea of applying sequential hypothesis testing techniques for DP auditing is novel and interesting
- The paper provides a practical test, with both non-trivial theoretical guarantees and good experimental results

Weakness:
- As stated multiple times in the paper, the main limitation of the test proposed is the assumption that the two neighbouring datasets $S$ and $S’$ are given as input. In most practical DP auditing settings, generating such pairs, especially worst-case ones, is the most challenging part, and this part is not addressed in the paper.

- The terms "black-box" and "white-box" are used inconsistently or imprecisely. For instance, in the first part, "black-box" meant that the auditor does not know the inner functioning of the mechanism $\mathcal{A}$, while in the DP-SGD auditing, “white-box” means that the auditor has access to the intermediate gradients, but still does not know the inner functioning of the mechanism $\mathcal{A}$. A clearer definition of these terms early in the paper would help.


- While the adaptation of the test from [40] is useful, the technical novelty compared to [40] is still unclear/incremental. The paper could do more to explicitly highlight what is technically new beyond switching from two-sided to one-sided.

---

> ### Author Rebuttal · Authors · 2025-07-31
>
> Thank you for the thoughtful review and helpful feedback. Below we address weaknesses and questions.
>
> # Weaknesses
>
> >`As stated multiple times in the paper, the main limitation of the test proposed is the assumption that the two neighbouring datasets S and S’ are given as input. In most practical DP auditing settings, generating such pairs, especially worst-case ones, is the most challenging part, and this part is not addressed in the paper.`
>
> We agree with the reviewer's observation and tried to make this point clear in our presentation. While it would be ideal to have an end-to-end universal auditing pipeline, finding these databases often depends on the specific mechanism and potential bugs being tested [A]. Indeed, the assumption that worst-case adjacent databases are known is a common practice in the literature. While existing work has explored various heuristics, such as online optimization and carefully crafted canaries [A, B], a more rigorous approach with worst-case guarantees remains a challenging open problem.
>
> >`The terms "black-box" and "white-box" are used inconsistently or imprecisely. For instance, in the first part, "black-box" meant that the auditor does not know the inner functioning of the mechanism, while in the DP-SGD auditing, “white-box” means that the auditor has access to the intermediate gradients, but still does not know the inner functioning of the mechanism. A clearer definition of these terms early in the paper would help.`
>
> We appreciate the feedback on our use of “black-box” and “white-box” terminology. We will ensure a precise and consistent use of these terms throughout the manuscript. Specifically, we define a black-box audit as one where the auditor observes only the mechanism's outputs, without knowledge of its internal functioning. Conversely, a white-box setting, like the DP-SGD auditing approach, provides access to specific internal states, such as intermediate gradients, allowing for more granular analysis.
>
> >`While the adaptation of the test from [40] is useful, the technical novelty compared to [40] is still unclear/incremental. The paper could do more to explicitly highlight what is technically new beyond switching from two-sided to one-sided.`
>
> See our response in the questions section below.
>
> # Questions
>
> >`What exactly are the “non-trivial adaptations and subtle but key modifications of the original results presented in [ 40 ]” (Line 140)? Could you explicitly enumerate or summarise what changes were needed to adapt the two-sided test to your one-sided setting?`
>
> Two-sided tests – like the hypothesis tested in [A] – are fundamentally different from one-sided tests, such as the one in our paper. We address this distinction in lines 176–215 in our manuscript, especially lines 188–198, which detail the necessary changes. In summary, the algorithmic changes are following:
>
> 1. First, we adapted the process $$\prod_t (1+ \lambda_t [f_t(X_t) - f_t(Y_t)])$$ from [A] by subtracting a term $\tau(\epsilon, \delta)$, which is derived in Theorem 3.1. This resulted in the new process $$\prod_t (1+ \lambda_t [f_t(X_t) - f_t(Y_t) - \tau(\epsilon, \delta)]).$$
>
> 2. Second, we restricted $\lambda$ to the interval $[0,1/(2+\tau(\epsilon,\delta))]$, while in [A] was considered to be in [-1/2,1/2]. This restriction is essential for two reasons:
>    - $\lambda \ge 0$ ensures that the process is a supermartingale under the null;
>    - and $\lambda \leq 1/(2+\tau(\epsilon,\delta))$ guarantees that the process remains nonnegative at all times.
>
>     Both of these are required for using Ville’s inequality for stopping-time guarantees.
>
> While seemingly minor, each of these changes is theoretically essential and not obvious a priori. Indeed, previous work attempting to adapt [A] to the one-sided setting did not provide stopping-time guarantees, and their corresponding process grew more slowly under the alternative than ours (see lines 199–215).
>
> A final difference from [A] is how we ensure fast growth under the alternative. While [A] proves that their process grows rapidly for some $\lambda^* \in [-1/2, 1/2]$, our analysis must guarantee that such a $\lambda^*$ exists within the more constrained interval we consider; this needs to be taken into account in our analysis (line 893).
>
> >`Improving further Theorem 3.1 via TV: In the proof of Theorem 3.1, it seems that you first upper bound the MMD by a Total Variation (TV). But it seems that Remark A.1.b) in the paper by Kairouz et al. (“The Composition Theorem for Differential Privacy”) provides tighter bounds on TV in terms of DP parameters. Can't you directly plug in those to yield an even tighter bound than your Theorem 3.1?`
>
> Thanks for the reference. Our goal was to connect MMD with Approximate DP, and in our proof we directly work with these two definitions. One could have alternatively first used a total variation (TV) bound on MMD, and continue by applying Remark A.1.b) from Kairouz et al. to the TV. We find that through both methods we arrive at the same MMD upper bounds. We will include this alternative proof in our final manuscript.
>
> >`Isn’t auditing DP-SGD in the white-box setting equivalent to auditing the mean mechanism applied to gradients, and thus section 4.2 is the same as section 4.1? If not, what are the differences?`
>
> We appreciate the opportunity to clarify the relationship between our empirical experiments in Section 4.1 (DP mean estimation) and Section 4.2 (DP-SGD). Indeed, we agree that our general auditing framework is similarly  applied to two sets of samples, mean estimation samples and white-box DP-SGD gradients, and more broadly, to observations from any two i.i.d. distributions.
>
> However, Section 4.2 demonstrates a crucial distinction: the overall procedure allows for auditing a model trained with DP without training multiple models as one would need according to black-box audits in section 4.1.
>
> Leveraging white-box access delivers significant efficiency gains for complex mechanisms like DP-SGD. Specifically, while black-box audits (as in Section 4.1) require training the full model for each new sample, white-box DP-SGD auditing avoids this by leveraging intermediate gradient updates. These gains apply both to batch and sequential testing. However, sequential testing can take even more advantage by further applying the auditing method sequentially, enabling DP-SGD auditing in less than one complete training run. This represents a substantial improvement in auditing efficiency for a complex, iterative mechanism like DP-SGD, differentiating it from a simpler, black-box mean mechanism audit.
>
>
> > `Is it possible to use your sequential test for auditing DP-SGD in the black-box model, i.e. you only have query access to the learned model and not access to gradients anymore?`
>
> Yes! it is possible to audit DP-SGD in a black-box setting where only query access to the learned model is available. In such a scenario, the auditing approach would effectively be the same as that presented in Section 4.1, as the entire training process would need to be run to draw each new sample for our sequential auditing framework. We will discuss this black-box DP-SGD auditing scenario and its implications in our experiments section for added clarity.
>
> ### References
>
> [A] Kong, William, et al. "DP-Auditorium: A Large-Scale Library for Auditing Differential Privacy." 2024 IEEE Symposium on Security and Privacy (SP). IEEE, 2024.
>
> [B] Nasr, Milad, et al. "Tight auditing of differentially private machine learning." USENIX Security Symposium. 2023.
>
> [H] Shekhar, Shubhanshu, and Aaditya Ramdas. "Nonparametric two-sample testing by betting." IEEE Transactions on Information Theory 70.2 (2023).

---

> > ### Comment · Reviewer_codM · 2025-08-04
> >
> > I would like to thank the authors for the rebuttal, which answers my questions. As noted by all other reviewers, the assumption that the two neighbouring datasets are given is the main limitation of this line of work. That is why I maintain my initial evaluation of the submission.

---

### Official Review · Reviewer_pxSP · 2025-07-03

**Clarity:** 2
**Significance:** 2
**Originality:** 3
**Rating:** 4
**Confidence:** 3

**Summary:**

The paper proposes a Maximum Mean Discrepancy (MMD)-based one-sided sequential test for auditing differential privacy. It enables anytime-valid inference with Type I error control, overcoming the inefficiency of prior batch tests. The authors also derive a new bound linking MMD to the Hockey-Stick divergence, which improves test power. Experiments show the method detects DP violations with far fewer samples—e.g., within a single DP-SGD training run, whereas the previous method requires a complete training round—thus making it practical and efficient for real-world use.

**Questions:**

It would be helpful if the authors could elaborate on how Algorithm 1 is applied to audit DP-SGD in practice. Since DP-SGD is an iterative algorithm where each update depends on previous model parameters, the gradients at different iterations are not i.i.d. samples from a fixed distribution pair. This seems to violate the assumption in Line 5 of Algorithm 1, where samples $ X_t \sim A(S) $ and $ Y_t \sim A(S')$ are assumed to be drawn independently from two fixed distributions.

**Ethical Concerns:**

["NO or VERY MINOR ethics concerns only"]

**Final Justification:**

I will keep my positive score.

**Limitations:**

yes

**Paper Formatting Concerns:**

No formatting concerns as far as I know.

**Quality:**

3

**Strengths And Weaknesses:**

**Strength:**
The paper presents a novel MMD-based one-sided sequential testing framework for auditing differential privacy, introducing testing-by-betting techniques that are new to this area. A notable strength is the ability to detect violations without requiring a full training run.


**Weakness:**
* The method assumes access to a fixed worst-case pair of neighboring datasets $(S, S')$. Identifying such pairs can be difficult in practice, limiting the method's applicability in fully black-box settings.
* The method relies on estimating the function $f_t$ via online optimization (e.g., Online Newton Step), which introduces additional computational overhead.
* It would also be helpful to include experiments comparing the accuracy of the proposed method with that of existing one-run auditing methods, to contextualize potential computation-accuracy tradeoffs.










* It would also be helpful to include experiment comparising the accuracy against one-run auditing methods

---

> ### Author Rebuttal · Authors · 2025-07-31
>
> Thank you for the thoughtful review and helpful feedback. Below we address weaknesses and questions.
>
> ## Weaknesses
> >`The method assumes access to a fixed worst-case pair of neighboring datasets. Identifying such pairs can be difficult in practice, limiting the method's applicability in fully black-box settings.`
>
> We appreciate the reviewer's observation. While it would be ideal to have an end-to-end universal auditing pipeline, finding these databases often depends on the specific mechanism and potential bugs being tested [A]. Indeed, the assumption that worst-case adjacent databases are known is a common practice in the literature. While existing work has explored various heuristics, such as online optimization and carefully crafted canaries [A, B], a more rigorous approach with worst-case guarantees remains a challenging open problem.
>
> Note that the applicability of the method is not necessarily limited by this: our method preserves its statistical validity when run on ${\it any}$ fixed pair of neighboring datasets, and even on time-varying pairs of datasets selected by heuristics. However, the power of the test might decrease when S and S' are not a worst-case pair of neighboring datasets.
>
> >`The method relies on estimating the function via online optimization (e.g., Online Newton Step), which introduces additional computational overhead.`
>
> The MMD-based two sample test in requires O(n^2) kernel evaluations the batch setting. Our sequential algorithm uses O(k) kernel evaluations in step k (see equation below line 973) which still adds up to O(n^2) kernel evaluations after n steps, but it often stops sooner, making the computational overhead minimal (if one exists). We discuss further implementation details of OGA in Section C.3.
>
> >`It would also be helpful to include experiments comparing the accuracy of the proposed method with that of existing one-run auditing methods, to contextualize potential computation-accuracy tradeoffs.`
>
> We thank the reviewer for this important question. First, it should be noted that the accuracy of the trained classifier using the DP-SGD algorithm is not affected by our auditing procedure following the white-box setting in [B].
>
> Regarding the hypothesis-test accuracy, our sequential testing framework naturally encompasses a one-run audit as a special case: when our sequential test does not reject the null hypothesis throughout the entire SGD training process, it effectively becomes equivalent to a one-run MMD-based audit using the complete gradient set. This ensures our method's accuracy is always at least as good as comparable one-run approaches while potentially allowing for faster detection rates.
>
> We thank the reviewer again for the opportunity to clarify this important point and will ensure the revised manuscript better positions our contribution relative to one-run auditing methods.
>
>
> # Questions
>
> >`It would be helpful if the authors could elaborate on how Algorithm 1 is applied to audit DP-SGD in practice.`
>
> This is an important point and we thank the opportunity to clarify this in detail. We will update the manuscript with this explanation.
>
> We adopt the standard white-box auditing procedure introduced as Algorithm 2 in [B]. This procedure gives the auditor access to all training hyperparameters (in particular, clipping norm and batch size for rescaling), online access to the intermediate privatized gradient during DP-SGD training, and the ability to insert canaries (specific crafted examples) to each sampled batch.
>
> The procedure works as follows: at  each training step $t$ a batch $B_t$ of examples is sampled. The auditor computes two private gradients: $\tilde{\nabla}[t]$ from the standard gradient $B_t$ and $\tilde{\nabla}'[t]$ where a canary gradient $g_t'$ was inserted. This canary is constructed to be orthogonal to all other per-example gradients within its batch. The auditor collects one-dimensional samples $x_t, y_t$, corresponding to the dot product between the canary $g_t'$ and the gradients: $x_t = \langle g_t',\tilde{\nabla}[t]\rangle$ and $y_t = \langle g_t',\tilde{\nabla}'[t]\rangle$.
>
> Since $\langle g_t',g\rangle = 0$ for all $g \in B_t$, then $\langle g_t',\tilde{\nabla}[t]\rangle =  \langle g_t',Z_t \rangle$ where $Z_t$ is the i.i.d. Gaussian noise added for privacy. Similarly, for the privatized gradient with the canary included we obtain that $\langle g_t',\tilde{\nabla}’[t]\rangle=\langle g_t',g_t’+\nabla[t] + Z_t’ \rangle=\|g_t’\|^2+\langle g_t',\nabla[t]\rangle + \langle g_t',Z_t’ \rangle = \|g_t’\|^2 + 0 +\langle g_t',Z_t’ \rangle$.
>
> Finally, the clipping norm and batch size are known by the auditor in the white-box setting, the samples can be rescaled. This process yields $\{x_t\}$ drawn from a Gaussian distribution $N(0,\sigma^2)$ and $\{y_t\}$ where the canary was present, drawn from $N(1, \sigma^2)$.
>
>
> >`Since DP-SGD is an iterative algorithm where each update depends on previous model parameters, the gradients at different iterations are not i.i.d. samples from a fixed distribution pair. This seems to violate the assumption in Line 5 of Algorithm 1, where samples Xt and Yt are assumed to be drawn independently from two fixed distributions.`
>
> The procedure above for auditing DP-SGD does satisfy the i.i.d assumption, given that it reduces to distinguishing i.i.d. samples from two Gaussian distributions. In practice, crafting time-varying canaries $g'_t$ to ensure exact orthogonality can be computationally intensive. For this reason, practitioners use a simpler static “Dirac” canary (all dimensions are set to zero and one element is set to 1). This well-established approximation is known to be highly effective and yields tight privacy bounds (see Fig. 14 in [B]). This method for generating samples for a two-sample test is widely adopted in the privacy auditing literature [A-G], but as the reviewer correctly pointed out, it does break the iid assumptions on the data. We will make sure to clarify this in the manuscript.
>
> However, we note that the proposed sequential auditing framework is amenable for time-varying distributions, for the two-sample test. See section 5.1. in [H].
>
> ### References
>
> Letters used for references ensure consistency accross reviewers.
>
> [A] Kong, William, et al. "DP-Auditorium: A Large-Scale Library for Auditing Differential Privacy." 2024 IEEE Symposium on Security and Privacy (SP). IEEE, 2024.
>
> [B] Nasr, Milad, et al. "Tight auditing of differentially private machine learning." USENIX Security Symposium. 2023.
>
> [E] Ponomareva, Natalia, et al. "How to dp-fy ml: A practical guide to machine learning with differential privacy." Journal of Artificial Intelligence Research (2023).
>
> [F] Steinke, Thomas, Milad Nasr, and Matthew Jagielski. "Privacy auditing with one (1) training run." NeurIPS (2023).
>
> [G] Andrew, Galen, et al. "One-shot empirical privacy estimation for federated learning." ICLR (2023).
>
> [H] Shekhar, Shubhanshu, and Aaditya Ramdas. "Nonparametric two-sample testing by betting." IEEE Transactions on Information Theory 70.2 (2023).

---

> ### Comment · Reviewer_pxSP · 2025-08-06
>
> Thanks for answering my question, I will keep my positive score.

---

### Official Review · Reviewer_7Q2v · 2025-07-23

**Clarity:** 3
**Significance:** 3
**Originality:** 3
**Rating:** 5
**Confidence:** 3

**Summary:**

This paper proposes new methods for auditing differentially private
algorithms. Specifically, the paper focuses on approximate
differential privacy and proposes to use a methodology inspired by
recent work on sequential hypothesis testing based on the theory of
e-values, which are roughly nonnegative random variables
characterizing the evidence for rejecting a null hypothesis, which
compose well adaptively. The proposed tests are based on maximum mean
discrepancy and uses Online Newton Step and Online Gradient Ascent for
learning the needed parameters. The paper presents an experimental
evaluation on multiple mechanisms and benchmarks.

**Questions:**

Your test uses OGA and ONS as black boxes. Does it means that you could obtain better tests by using similar algorithms with improved guarantees?

**Ethical Concerns:**

["NO or VERY MINOR ethics concerns only"]

**Limitations:**

yes

**Quality:**

3

**Strengths And Weaknesses:**

Strengths

- The paper uses a recently developed approach to sequential hypothesis
  testing to address the problem of auditing differentially private
  mechanisms

- the tests presented in the paper improves considerably in terms of
  performance with respect to previous approaches

Weaknesses

- The experimental evaluation does not provide any direct comparison with
  previous methods,

- similar to some previous work, the paper assumes that worst-case
  adjacent databases are known.

Several work have studied auditing methods for differentially private
mechanism, especially in the context of differentially private
optimization and learning. This paper brings new methods to this area
by introducing an approach based on sequential hypothesis testing
using e-values. The paper combines this ideas in a test using off the
shelf learning algorithms to learn some of the parameters needed for
deciding the test. The proposed tests are more adaptive than the ones
proposed in previous work and are able to converge faster. This makes
them more efficient for auditing purposes than previous
approaches. The efficiency is evaluated experimentally in the paper
with encouraging results. Although, to be fair, there is not concrete
experimental comparison of the proposed method with previous works.
One limitation, acknowledged by the paper, is that the paper assumes
the knowledge of worst-case adjacent databases. This makes the
proposed approach more aligned with tools for auditing privacy
parameters rather than with tools for finding privacy violations.

---

> ### Author Rebuttal · Authors · 2025-07-31
>
> Thank you for the thoughtful review and helpful feedback. Below we address weaknesses and questions.
>
> # Weaknesses
>
> >`The experimental evaluation does not provide any direct comparison with previous methods (...)  The efficiency is evaluated experimentally in the paper with encouraging results. Although, to be fair, there is not concrete experimental comparison of the proposed method with previous works.`
>
>
> In lines 287-290 we briefly compare our results to those of DP-Auditorium [A] in the black box setting. We will add this comparison more explicitly to our result tables.
>
> The experiment conducted to audit DP-SGD in section 4.2 follows the auditing procedure from [B]. We will add a comparison to the lower bounds that they report in their Figure 3.
>
>
> >`One limitation, acknowledged by the paper, is that the paper assumes the knowledge of worst-case adjacent databases.`
>
> We appreciate the reviewer's observation. While it would be ideal to have an end-to-end universal auditing pipeline, finding these databases often depends on the specific mechanism and potential bugs being tested [A]. Indeed, the assumption that worst-case adjacent databases are known is a common practice in the literature. While existing work has explored various heuristics, such as online optimization and carefully crafted canaries [A, B], a more rigorous approach with worst-case guarantees remains a challenging open problem.
>
> # Questions
> >`Your test uses OGA and ONS as black boxes. Does it means that you could obtain better tests by using similar algorithms with improved guarantees?`
>
> Great question! The answer is yes: the faster the stochastic process considered for testing grows under the alternative, the sooner the test rejects.
>
> Indeed, two recent papers proposed to replace ONS by algorithms that allow optimizing the parameters $\lambda$ over a larger interval and with smaller regret constants: [C] proposed to use the universal portfolio and [D] an optimistic interior point method. We empirically verified the practical benefits of using the approach from [C] and will include these new experiments in the appendix.
>
> ### References
>
> [A] Kong, William, et al. "DP-Auditorium: A Large-Scale Library for Auditing Differential Privacy." 2024 IEEE Symposium on Security and Privacy (SP). IEEE, 2024.
>
> [B] Nasr, Milad, et al. "Tight auditing of differentially private machine learning." 32nd USENIX Security Symposium (USENIX Security 23). 2023.
>
> [C] Waudby-Smith, Ian, Ricardo Sandoval, and Michael I. Jordan. "Universal log-optimality for general classes of e-processes and sequential hypothesis tests." arXiv preprint arXiv:2504.02818 (2025).
>
> [D] Chen, Can, and Jun-Kun Wang. "Optimistic interior point methods for sequential hypothesis testing by betting." arXiv preprint arXiv:2502.07774 (2025).

---

> > ### Comment · Reviewer_7Q2v · 2025-08-04
> >
> > Thanks for your response. It would be great if you could integrate more the comparison with DP-auditorium into the evaluation tables.

---

### Decision · Program_Chairs · 2025-09-17

**Decision:**

Accept (poster)

**Comment:**

This paper proposes a sequential testing framework for auditing DP.

All the reviewers agree the method is novel, and the results are very promising. Although some reviewers concern about the assumption of the neighbouring datasets, they believe the results are strong enough.

Therefore, I recommend the acceptance of the paper.